# Postsynaptic cell type and synaptic distance do not determine efficiency of monosynaptic rabies virus spread measured at synaptic resolution

**Maribel Patiño[1,2,3], Willian N Lagos[1], Neelakshi S Patne[1], Paula A Miyazaki[1], Sai Krishna Bhamidipati[1], Forrest Collman[4], Edward M Callaway[1]\***

[1]Systems Neurobiology Laboratories, The Salk Institute for Biological Studies, La Jolla, United States; [2]Neuroscience Graduate Program, University of California, San Diego, La Jolla, United States; [3]Medical Scientist Training Program, University of California, San Diego, La Jolla, United States; [4]Allen Institute for Brain Science, Seattle, United States

**Abstract** Retrograde monosynaptic tracing using glycoprotein-deleted rabies virus is an important component of the toolkit for investigation of neural circuit structure and connectivity. It allows for the identification of first-order presynaptic connections to cell populations of interest across both the central and peripheral nervous system, helping to decipher the complex connectivity patterns of neural networks that give rise to brain function. Despite its utility, the factors that influence the probability of transsynaptic rabies spread are not well understood. While it is well established that expression levels of rabies glycoprotein used to trans-complement G-deleted rabies can result in large changes in numbers of inputs labeled per starter cell (convergence index [CI]), it is not known how typical values of CI relate to the proportions of synaptic contacts or input neurons labeled. And it is not known whether inputs to different cell types, or synaptic contacts that are more proximal or distal to the cell body, are labeled with different probabilities. Here, we use a new rabies virus construct that allows for the simultaneous labeling of pre- and postsynaptic specializations to quantify the proportion of synaptic contacts labeled in mouse primary visual cortex. We demonstrate that with typical conditions about 40% of first-order presynaptic excitatory synapses to cortical excitatory and inhibitory neurons are labeled. We show that using matched tracing conditions there are similar proportions of labeled contacts onto L4 excitatory pyramidal, somatostatin (Sst) inhibitory, and vasoactive intestinal peptide (Vip) starter cell types. Furthermore, we find no difference in the proportions of labeled excitatory contacts onto postsynaptic sites at different subcellular locations.

**\*For correspondence:**
callaway@salk.edu

## Editor's evaluation

This study provides valuable new information on using monosynaptic rabies virus for trans-synaptic tracing, a widely used tool to map synaptic inputs to specific types of neurons in the mammalian brain. The authors provide convincing quantitative information about the efficiency for the rabies virus to cross synapses of visual cortical neurons in the mouse. They further demonstrate that the cortical cell type and location of synapse in the postsynaptic cell do not appear to affect the efficiency for the rabies virus to cross synapses.

## Introduction

Monosynaptic rabies tracing using glycoprotein (G)-deleted rabies virus (RVdG) is a powerful tool for the study of neural circuit connectivity. This method enables scientists to label, genetically manipulate, or monitor the activity of brain-wide monosynaptic inputs to cell populations of interest. Since its introduction (*Wickersham et al., 2007a*; *Wickersham et al., 2007b*) it has been widely used for the identification of presynaptic inputs to single neurons (*Marshel et al., 2010*; *Rancz et al., 2011*; *Rossi et al., 2020*; *Wertz et al., 2015*), projection-defined neurons (*Cruz-Martín et al., 2014*; *Levine et al., 2014*), adult-born neurons (*Deshpande et al., 2013*; *Garcia et al., 2014*), transplanted neurons (*Doerr et al., 2017*; *Grealish et al., 2015*), hPSC-derived organoid neurons (*Andersen et al., 2020*; *Miura et al., 2020*), and genetically defined excitatory neurons (*DeNardo et al., 2015*; *Kim et al., 2015*), inhibitory neurons (*Miyamichi et al., 2013*; *Wall et al., 2016*), and non-neuronal cell types (*Clark et al., 2021*; *Mount et al., 2019*). In addition to being used for the identification of inputs, the incorporation of $Ca^{2+}$ indicators and light-activated opsins into rabies reagents (*Osakada et al., 2011*) allows rabies tracing experiments to probe the relationship between function and connectivity (*Rossi et al., 2020*; *Tian et al., 2016*; *Wertz et al., 2015*; *Wester et al., 2019*). The ability of genetically modified RVdG to selectively spread retrogradely between synaptically connected cells allows for the identification of presynaptic partners regardless of their distance from one another and has led to novel insights throughout the nervous system.

Despite the utility and widespread use of monosynaptic rabies tracing to study neural connectivity, there is uncertainty about the efficiency of transsynaptic spread from starter cells to input neurons and what factors influence the probability of spread. Although studies quantifying inputs to single neurons have provided some insight into the efficiency of spread (*Marshel et al., 2010*; *Miyamichi et al., 2011*; *Rancz et al., 2011*; *Wertz et al., 2015*) results vary widely across experimental conditions and no direct measurements of spread efficiency are available. Furthermore, using the convergence index (CI), defined as the number of rabies-labeled input neurons divided by the number of starter cells, recent studies have shown that the number of input neurons labeled per starter cell can be changed by as much as 10-fold by modifying the rabies glycoprotein (*Kim et al., 2016*), replacing the rabies virus strain (*Reardon et al., 2016*), or adjusting the level of glycoprotein expression (*Lavin et al., 2020*). Despite allowing quantitative comparisons between different reagents and conditions, these studies relied on methods that result in large animal to animal variability, require tedious counting across many animals, and can display high variability depending on the number of starter neurons (*Tran-Van-Minh et al., 2022*). Most importantly, CI measurements fail to quantify what proportion of all inputs are labeled. Additionally, the use of CI fails to address questions of the influence of biological factors on rabies spread efficiency, such as distance of synapses to the starter cell soma or differences of spread from different starter cell types.

In this study we examine the efficiency of RVdG retrograde spread from starter cell to input neurons at the synaptic level. We designed a new genetically modified rabies virus that labels presynaptic terminals with synaptophysin-RFP (SynPhRFP) and excitatory postsynaptic densities with postsynaptic density-95-GFP (PSD95GFP). Because more than 99% of PSD95 postsynaptic puncta colocalize with a presynaptic terminal (*Micheva et al., 2010*), this construct allows us to quantify the proportion of excitatory synapses on a starter cell that have their corresponding input neuron labeled with rabies virus (defined here as synaptic fraction [SF]). We find that with the particular reagents and conditions that we used, rabies retrogradely labels about 35–40% of excitatory synaptic contacts onto each of the excitatory and inhibitory cell populations we tested. Furthermore, we found that SF does not vary by the proximity or distance of synapses to the starter cell soma or by the type of neuronal dendrite. Finally, we model the factors that influence the relationship between SF and the fraction of input neurons labeled (input fraction [IF]) and show that under plausible conditions for our results in mouse visual cortex the IF/SF ratio is about 0.75, so an SF of 40% corresponds to an IF of 30% of input neurons labeled. This analysis also allows an estimate of the probability of rabies spread across a single synaptic contact (unitary synaptic efficiency, U) which we estimate to be about 0.28 (28%) when SF = 0.4. Overall, this study provides insight into some long-standing questions about the efficiency of monosynaptic rabies tracing. We further discuss that desired tracing efficiency varies depending on experimental aims, the likely relationships between SF and proportion of input neurons that are labeled, and that efficiency was not maximized with the experimental design used here.

## Results

### RVdG-PSD95GFP-SynPhRFP viral construct allows simultaneous fluorescent labeling of pre- and postsynaptic specializations

To investigate the transsynaptic spread of rabies virus from starter cells to input neurons at synaptic resolution, we developed a high-throughput and precise method to directly identify synapses labeled by transsynaptic spread. We created a new deletion-mutant rabies virus construct that expresses two synaptic fusion protein transgenes from the rabies G locus (*Figure 1A*), modeled after existing rabies constructs known to label multiple subcellular compartments of neurons (*Wickersham et al., 2013*). One gene encodes for a fusion protein of the excitatory postsynaptic marker, postsynaptic density 95 (PSD-95), and enhanced green fluorescent protein (eGFP). The other encodes for a fusion protein of the presynaptic marker synaptophysin (SynPh) and the bright and photostable red fluorescent protein TagRFP-T (*Shaner et al., 2008*). We chose the synaptophysin-TagRFP-T (SynPhRFP) fusion protein as it has previously been shown to result in a punctate red fluorescent pattern that colocalizes with varicosities on axons when expressed from the rabies genome (*Wickersham et al., 2013*). Injection of SAD-B19 EnvA+ RVdG-PSD95GFP-SynPhRFP in primary visual cortex (V1) of *Sim1*^Cre mice expressing TVA and oG in Cre+ layer 5 neurons (derived from adeno-associated viruses [AAV] helper viruses, see Materials and methods) resulted in strong punctate green and red fluorescent labeling (*Figure 1B*), with green labeling prominent on dendrites and dendritic spines (*Figure 2E and F*). Super-resolution Airyscan imaging (see Materials and methods) revealed that GFP labeling colocalized with a subset of the puncta labeled with antibody staining against endogenous PSD-95 (*Figure 1C*). In addition to synaptic labeling there were typically nuclear aggregates of both red and green fluorescent protein in infected neurons (*Figure 1B*).

### Efficiency of rabies transsynaptic spread across excitatory synapses onto excitatory starter neurons

To quantify the proportion of synaptic contacts labeled with rabies virus, we injected a mixture of three Cre-dependent helper AAVs into the primary visual cortex (V1) of Nr5a1-Cre mice to target initial rabies infection to a sparse population of L4 excitatory neurons (*Harris et al., 2014*; *Figure 3A, B*). (1) AAV8-nef-AO-66/71-TVA950 (AAV-CIAO-TVA) expresses the TVA receptor for the avian sarcoma leukosis virus envelope protein, EnvA, which is necessary for entry of pseudotyped EnvA+ RVdG into Cre+ cells. However, recombinant-independent off-target leak expression of transgenes is common in recombinase-dependent DIO and FLEX AAV constructs. Because even miniscule quantities of TVA leak expression is sufficient for pseudotyped rabies to enter off-target cells and confound results (*Callaway and Luo, 2015*), it was necessary to minimize leak expression of TVA. We therefore used the novel cross-over insensitive ATG-out (CIAO) AAV construct, which has been shown to nearly eliminate leak expression and provide reliable and targeted transgene expression (*Fischer et al., 2019*), to express TVA in our population of interest. CIAO constructs have the ATG codon placed outside of the loxp mutant pairs loxp66/71 sites to ensure the gene coding region is out of frame with the ATG start signal in the absence of Cre-mediated recombination. Experiments in which AAV-CIAO-TVA and EnvA+ RVdG were injected into V1 of Cre-negative mice resulted in 15.83±11.33 rabies-infected neurons (mean ± SEM, n=3 mice) indicating low levels of Cre-independent TVA expression. (2) AAV8-hSyn-FLEX-H2BBFP-oG expresses optimized rabies glycoprotein (oG) (*Kim et al., 2016*), which allows for trans-complementation in EnvA+ RVdG-infected neurons, also termed starter cells, allowing the rabies to spread retrogradely into presynaptically connected inputs. Co-expression of nuclear mTagBFP2 (BFP), a brighter and more photostable blue fluorescent protein (*Subach et al., 2008*), and oG allows for the unambiguous identification of starter cells (*Figure 3C and E*), defined as any cell that expresses both oG and rabies transgenes. (3) Because H2BBFP only labels the nucleus of starter cells, we also used AAV-CAG-FLEX-smFP_myc to generate cytoplasmic labeling, allowing dendrites of starter cells to be traced. This AAV expresses spaghetti monster fluorescent protein (smFP), which consists of dark non-fluorescent GFP fused to 10 Myc epitope tags that can be combined with anti-Myc antibody staining to attain bright fluorescent labeling of subcellular structures (*Viswanathan et al., 2015*). Infection of L4 excitatory neurons with AAV-CAG-FLEX-smFP_myc resulted in strong dendritic labeling that improved the ability to accurately trace apical and basal dendrites of starter cells (*Figure 3C, E, and F*). Three weeks after injection of a mixture of the three AAV helper

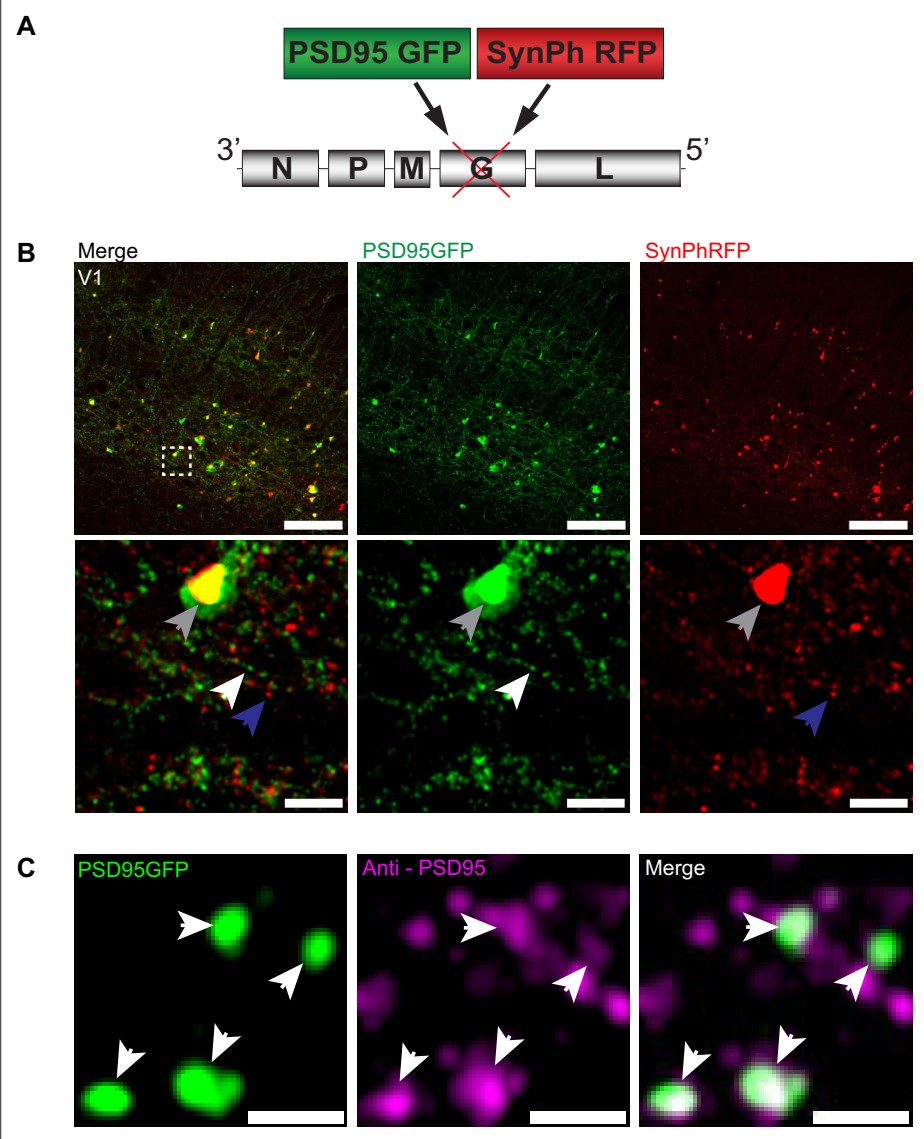

**Figure 1.** RVdG-PSD95GFP-SynPhRFP allows simultaneous fluorescent labeling of pre- and postsynaptic densities. (**A**) Schematic of RVdG-PSD95GFP-SynPhRFP viral construct design. Two transgenes were inserted into the G locus of the rabies genome. One encodes a presynaptically targeted fluorescent fusion protein, synaptophysin TagRFP-T (SynPhRFP), and the other a postsynaptically targeted fluorescent fusion protein, PSD-95 eGFP (PSD95GFP). (**B**) Coronal sections of *Sim1*[Cre] mouse expressing TVA and oG in V1 infected with EnvA+ RVdG-PSD95GFP-SynPhRFP imaged at 20× with confocal microscopy. Top row shows neurons expressing both PSD95GFP and SynPhRFP fusion proteins. Bottom row shows a zoomed in max intensity projection reconstructed image of the region enclosed by the dashed square in the top row. White arrows point to PSD-95 puncta, blue arrows to synaptophysin puncta, and gray arrows to large non-specific nuclear fluorescent aggregates. Scale bars represent 100 μm (top row) or 10 μm (bottom row). (**C**) Airyscan super-resolution max intensity projection reconstructed images taken at 63× showing colocalization of PSD95GFP fusion protein expressed from the rabies genome with anti-PSD95 antibody staining in magenta. Scale bar = 1 μm.

The online version of this article includes the following source data for figure 1:

**Source data 1.** Plasmid sequence for RVdG-PSD95GFP-SynPhRFP construct.

viruses, EnvA+ RVdG-PSD95GFP-SynPhRFP was injected into the same location in V1 and allowed to express for 7 days. As expected from transsynaptic spread, we observed that many thalamic dLGN long-distance inputs to L4 excitatory neurons were reliably labeled with RVdG-PSD95GFP-SynPhRFP (*Figure 3D*).

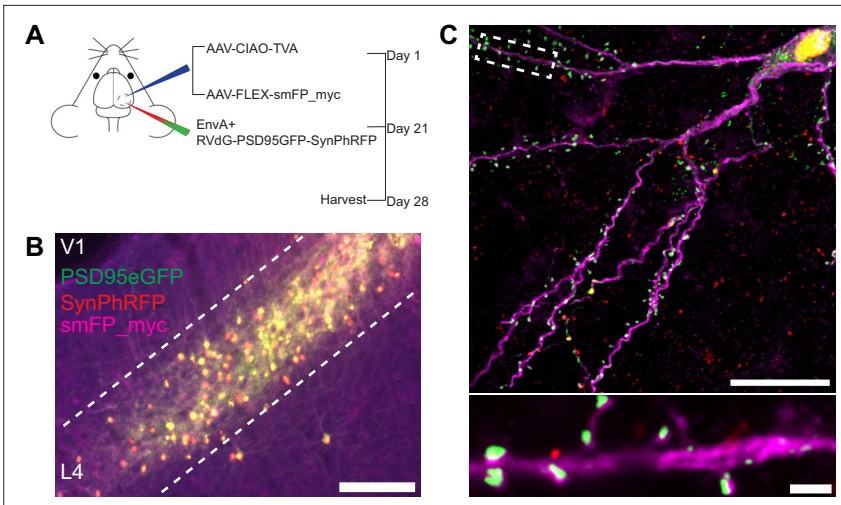

**Figure 2.** Experimental design for control experiments omitting oG. (**A**) Schematic illustration of experimental design and timeline for control experiments without glycoprotein. *Nr5a1*[Cre] mice were injected in V1 with a mixture of AAV-CIAO-TVA and AAV-FLEX-smFP-myc. Three weeks later EnvA+ RVdG-PSD95GFP-SynPhRFP was injected into the same site and allowed to express for 7 days. (**B**) Representative image of V1 injection site, obtained using widefield fluorescence microscopy at 10×. Scale bar = 200 µm. (**C**) Max intensity projection reconstruction of images obtained using Airyscan super-resolution imaging at 63×. Top, example image of rabies-infected neuron labeled with smFP_myc, without glycoprotein. Bottom, zoomed in image of boxed region in top image, illustrating PSD-95 puncta colocalized with cytoplasmic smFP_myc. Scale bar = 20 µm (top) and scale bar = 2 µm (bottom).

To quantify the proportion of excitatory synaptic contacts labeled on starter neurons (defined as SF), we first identified starter neurons that also expressed smFP using widefield fluorescence microscopy. Select areas of starter neuron dendrites that could be traced back to their parent cell bodies were imaged using Airyscan super-resolution microscopy (*Huff et al., 2017*) to increase the resolution and signal-to-noise and allow visualization of single synaptic puncta (*Figure 3F*). To assess the transsynaptic spreading efficiency, we quantified the proportion of starter cell excitatory postsynaptic specializations, labeled with PSD95GFP, that were directly apposed with rabies-labeled presynaptic terminals, labeled with SynPhRFP (*Figure 3F*). Because L4 excitatory neurons connect to one another, the direct connections between starter cells can generate co-labeling of pre- and postsynaptic specializations independent from transsynaptic spread. Co-labeling independent of transsynaptic spread might also result from direct rabies infection of the small numbers of non-starter neurons infected with rabies due to leak expression of TVA (see above). It was therefore important to begin by quantifying how much of this 'background label' is present under conditions in which similar numbers of L4 neurons are directly infected with EnvA+ RVdG-PSD95GFP-SynPhRFP, but there is no transsynaptic spread. To quantify this we calculated the proportion of PSD95GFP puncta apposed to SynPhRFP in experiments that omitted oG (no AAV8-hSyn-FLEX-H2BBFP-oG, *Figure 2A, B, and C*). The omission of oG prevents retrograde spread of rabies virus and labeling of input neurons (*Figure 3D*), therefore observed SynPhRFP colocalization with PSD95GFP must be a result of L4 to L4 excitatory starter neuron connections and/or infection via TVA leak expression. We found that neurons in which oG was used for trans-complementation displayed significantly higher proportions of PSD95GFP puncta colocalized with SynPhRFP compared to neurons in which oG was omitted (50.61 ± 3.92% vs 8.86 ± 1.79% mean ± SEM respectively, Wilcoxon rank-sum test, p=0.001, n=9 neurons across 3 mice and n=5 neurons across 2 mice; *Figure 3G*). Within experimental condition groups, puncta colocalization did not vary significantly across neurons from distinct experimental animals. The difference between these values (50.61–8.86%) yields an estimate of 42% of postsynaptic densities whose presynaptic partners are transsynaptically labeled (SF = 0.42).

To determine how SF is related to distance from the cell body we sampled from different portions of the dendritic arbors of L4 starter cells. Sampled areas within 75 µm of the soma were classified as proximal dendritic domains and areas within 75 µm of the pial surface were considered distal domains. These distal domains are estimated to be about 250–350 µm from the soma. We found no significant

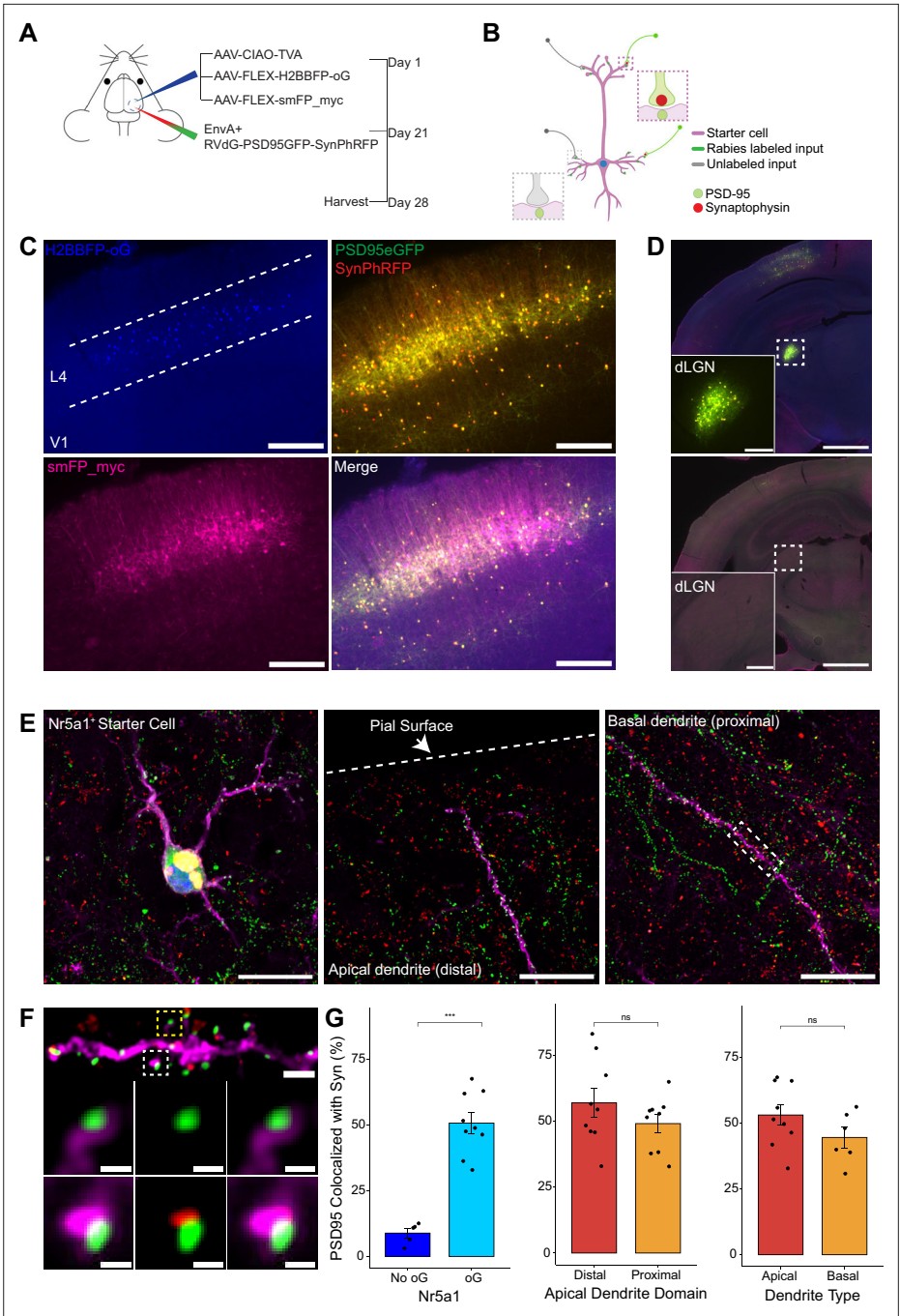

**Figure 3.** Efficiency of transsynaptic spread from excitatory L4 Nr5a1+ starter cells to excitatory inputs.
(**A**) Schematic illustration of experimental design and timeline for monosynaptic rabies tracing. *Nr5a1*[Cre] mice were injected in V1 with a mixture of AAV-CIAO-TVA, AAV-FLEX-H2BBFP-oG, and AAV-FLEX-smFP_myc. Three weeks later EnvA+ RVdG-PSD95GFP-SynPhRFP was injected into the same site and allowed to express for 7 days.
(**B**) Schematic of rabies retrograde spread efficiency quantification paradigm. Starter neurons are distinguished from input neurons based on expression of nuclear BFP from AAV-FLEX-H2BBFP-oG in addition to fusion proteins from EnvA+ RVdG-PSD95GFP-SynPhRFP. Starter neurons expressing smFP_myc are used for synaptic quantification to allow tracing of distal dendrites. Synaptic fraction is measured by quantifying the proportion of postsynaptic densities (PSD95GFP) on the starter neuron apposed with rabies-labeled presynaptic terminals (SynPhRFP). (**C**) Representative example images of V1 injection site, obtained using widefield fluorescence microscopy at ×10 magnification. Scale bar = 200 μm. (**D**) Coronal section example images obtained using widefield fluorescence microscopy at 10× showing long-range monosynaptic input neurons in dorsal lateral

*Figure 3 continued on next page*

*Figure 3 continued*

geniculate nucleus (dLGN) to Nr5a1+ L4 neurons in V1 when using the new RVdG construct (top). No retrograde spread is observed when glycoprotein is omitted, see *Figure 2* for additional information. Insets are zoomed in images of dashed box regions. Scale bar represents 1 mm in hemisection image or 200 μm in inset. (**E**) Max intensity projection reconstruction of images obtained using Airyscan super-resolution imaging at 63×. Left, example image of starter neuron (H2BBFP+,PSD95GFP+, and SynPhRFP+) labeled with smFP_myc. Middle, example image of the distal domain of an apical dendrite of a starter neuron. Right, example image of the proximal domain of a basal dendrite. Scale bar = 20 μm (all three). (**F**) Spatial resolution using Airyscan imaging is sufficient to quantify rabies transsynaptic spread at the synaptic level. Zoomed in max intensity projection reconstructed image of boxed region in (**E**) right, illustrating PSD-95 puncta colocalized with cytoplasmic smFP_myc. Top, yellow boxed region highlights a spine with PSD-95 puncta without an apposed rabies-labeled presynaptic density. White boxed region highlights a spine with PSD-95 puncta with an apposed rabies-labeled presynaptic density. Middle row, zoomed in max intensity projection reconstructed images of yellow boxed region and bottom rows are zoomed in max intensity projection reconstructed images of white boxed region. Top, scale bar = 2 μm and middle and bottom scale bar = 0.5 μm. (**G**) Percent of postsynaptic densities (PSD95GFP) on Nr5a1+ starter cells apposed with rabies-labeled presynaptic terminals (SynPhRFP). Left, quantification of colocalization at baseline (no glycoprotein) due to L4 to L4 connections compared to colocalization from transsynaptic spread (with glycoprotein). Middle, colocalization on the distal vs proximal domains of apical dendrites. Right, colocalization on apical vs basal dendrites. Values are reported as mean ± SEM. Statistics were calculated from the Wilcoxon rank-sum test for non-parametric comparisons. Individual data points (circles) indicate values for each neuron. n (number of neurons) = 5 and 9 and N (number of mice) = 2 and 3 for no oG and oG groups respectively. p-value > 0.05 = not significant (ns).

differences between SF at proximal regions of apical dendrites compared to the distal regions as determined by PSD95GFP and SynPhRFP puncta colocalization (49.14 ± 3.49% vs 57 ± 5.43% respectively, Wilcoxon rank-sum test, p=0.34; n=9 neurons across 3 mice, *Figure 3G*). We also compared SF at basal dendrites versus apical dendrites and observed no significant difference (44.52 ± 3.96% vs 53.05 ± 3.99% respectively, Wilcoxon rank-sum test, p=0.22; n=9 neurons across 3 mice and n=6 neurons across 3 mice, *Figure 3G*). Note that a fraction of the observed synaptic labeling might occur through indirect pathways rather than direct spread of rabies virus at the observed synapse (see below). This fraction is small relative to the variability in our measured SF at distal versus proximal or apical versus basal dendrites.

## Efficiency of rabies transsynaptic spread across excitatory synapses onto inhibitory starter neurons

To assess possible differences in efficiency of rabies retrograde spread for different starter cell types, we conducted monosynaptic rabies tracing using RVdG-PSD95GFP-SynPhRFP as described above, but using mouse lines that express Cre recombinase in two distinct classes of inhibitory neurons. We used the knock-in *Sst*^Cre and *Vip*^Cre lines to target initial infection to either somatostatin (Sst)-expressing or vasoactive intestinal peptide (Vip)-expressing inhibitory neurons (*Figure 4A*). We selected Sst and Vip interneurons as both exhibit very low levels of recurrent connections compared to parvalbumin-expressing neurons, which connect extensively to each other (*Campagnola et al., 2022*; *Pfeffer et al., 2013*). We found no significant difference between the percent of PSD95GFP puncta apposed to SynPhRFP on Sst dendrites compared to Vip dendrites (36.13 ± 1.73% vs 38.38± 1.22% respectively, Wilcoxon rank-sum test, p=0.44, n=9 neurons across 3 mice per line; *Figure 4F*). These values are comparable to the SF for labeling inputs to L4 starter cells (42%, see above). We also assessed whether SF varied based on distance from the cell body. For Sst starter cells, colocalization of PSD95GFP with SynPhRFP on proximal dendritic domains (within 75 μm of the cell body) did not differ significantly from distal dendritic domains (within 75 μm of the pial surface) (37.50 ± 1.93% vs 33.85 ± 2.43% respectively, Wilcoxon rank-sum test, p=0.55, *Figure 4F*). We observed similar results for Vip starter cells, with no difference between SF at proximal versus distal domains (39.80 ± 1.90% vs 36.122 ± 1.14% respectively, Wilcoxon rank-sum test, p=0.11, *Figure 4F*).

## Relationships between SF, IF, and U

Because prior studies have found large variability in the number of rabies-labeled input cells per starter cell (CI), there is great interest in understanding what fraction of all input neurons are labeled

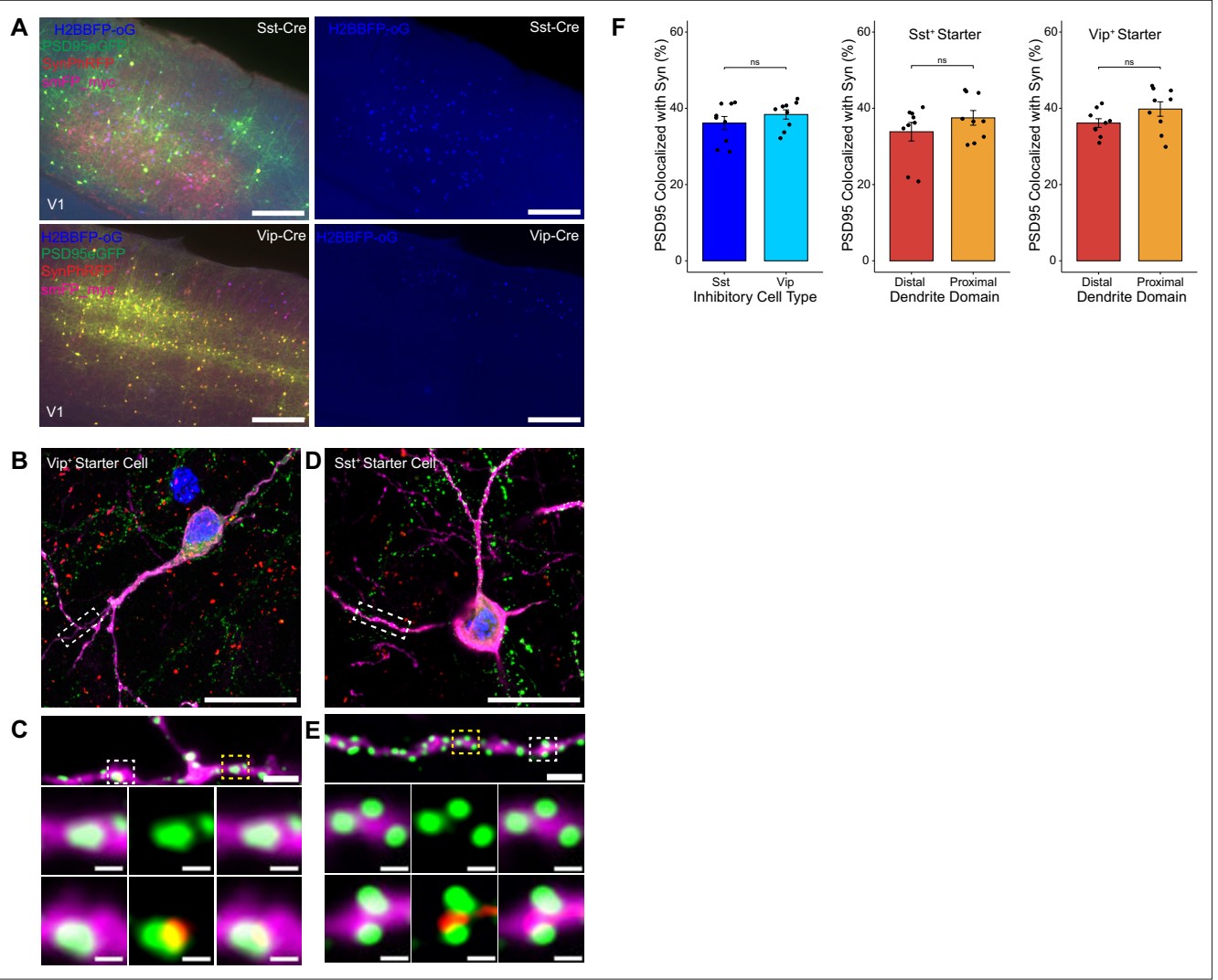

**Figure 4.** Efficiency of transsynaptic spread from inhibitory starter cells to excitatory inputs. (**A**) Representative example images of V1 injection site for *Sst*^Cre mouse line (top) and *Vip*^Cre mouse line (bottom), obtained using widefield fluorescence microscopy at 10×. Scale bar = 200 μm. (**B**–**E**) Max intensity projection reconstruction of images obtained using Airyscan super-resolution imaging at 63×. Example image of Vip+ (**B**) and Sst+ (**D**) starter neurons (H2BBFP+, PSD95GFP+, and SynPhRFP+) labeled with smFP_myc. Scale bar = 20 μm. (**C** and **E**) Zoomed in images of boxed regions in (**B** and **D**) respectively, illustrating PSD-95 puncta colocalized with cytoplasmic smFP_myc. Top, yellow boxed region highlights a spine with PSD-95 puncta without an apposed rabies-labeled presynaptic density. White boxed region highlights a spine with PSD-95 puncta with an apposed rabies-labeled presynaptic density. Middle row, zoomed in images of yellow boxed region and bottom rows are zoomed in images of white boxed region. Top, scale bar = 2 μm and middle and bottom scale bar = 0.5 μm. (**F**) Left, percent of postsynaptic densities (PSD95GFP) on Vip+ and Sst+ starter cells apposed with rabies-labeled presynaptic terminals (SynPhRFP). Middle, colocalization on the distal vs proximal domains of Sst+ dendrites. Right, colocalization on the distal vs proximal domains of Vip+ dendrites. Values are reported as mean ± SEM. Statistics were calculated from Wilcoxon rank-sum test for non-parametric comparisons. Individual data points (circles) indicate values for each neuron. n (number of neurons) = 9 and N (number of mice) = 3. p-value > 0.05 = not significant (ns).

(IF). But for the great majority of circuits, the actual numbers of inputs to each neuron is unknown, so it is not possible to calculate IF from CI. Below, along with detailed derivation in Materials and methods, we provide a formal analysis and quantitative estimates of the relationships between SF and IF for cortical starter neurons. Overall, these analyses indicate that the SF data we have presented here provide a good estimate and important new insight into IF under the tracing conditions that we have used. But it is important to consider several factors that make SF likely to be somewhat larger than IF. Our analyses (*Figures 5–6*, and more detailed considerations in Materials and methods) indicate that for the cortical circuits we studied, our observed SF value of around 40% likely corresponds to an IF of about 30%. We further estimate that when SF is 40% for a cortical neuron, the probability of

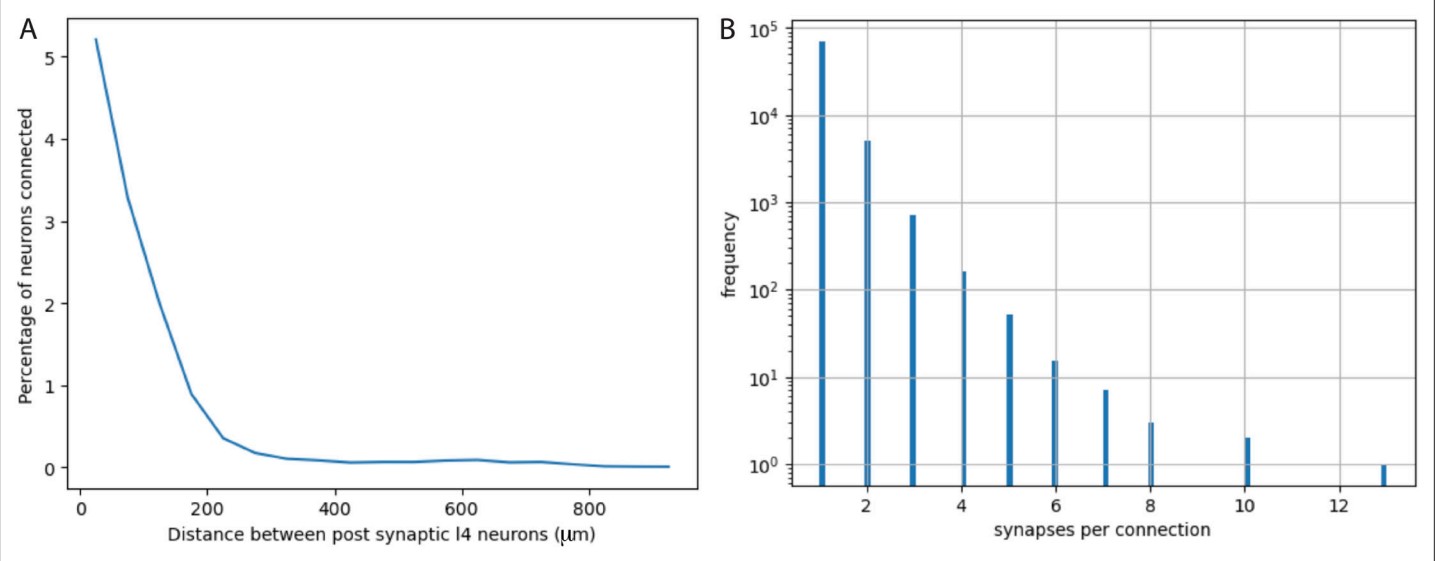

**Figure 5.** Quantification from electron micrographic reconstructions of: (**A**) shared inputs between layer 4 excitatory neurons and (**B**) the number of synapses per excitatory neuron to excitatory neuron connection. (**A**) The percentage of inputs to a layer 4 excitatory neuron that provide common input to another layer 4 excitatory neuron (percentage of neurons connected) at increasing distances between layer 4 neurons. Based on identification of 17,883 presynaptic inputs to layer 4 cells. (**B**) Distribution of the numbers of synapses per connection for 76,678 excitatory neuron to excitatory neuron connections identified in electron micrographic reconstructions of a volume from adult mouse visual cortex. Y-axis is plotted on a log scale. Average number of synapses per connection = 1.0965 ± 0.3726 (STD). The overwhelming majority of connections involve only one synaptic contact.

rabies spread across a single synaptic contact (unitary synaptic efficiency, U) is likely to be about 28%. For most applications, this spreading efficiency is likely to be near optimal because it can allow for detection of weak versus strong input sources (see Discussion); but there is clearly room for increasing spread if the experimental application or question calls for labeling closer to all input neurons.

What is the expected relationship between SF and IF and what are the factors that make SF likely to be larger than IF? We have defined SF as the proportion of synaptic contacts that are labeled on a given starter neuron. It is important to note that SF will typically not be the same as the proportion of input neurons that are labeled (IF). In Materials and methods along with *Figures 5–6*, we formally consider factors that are likely to impact the relationship between SF and IF. We consider these relationships in the context of cortical circuits, as we have studied here, but the factors considered can also be applied to other brain areas with different circuit organization.

There are two main features of cortical circuit organization that influence the SF to IF relationship. These are: (1) the extent to which different starter neurons share common inputs; and (2) the distribution of the numbers of synaptic contacts provided per input neuron onto a particular starter neuron. Both of these factors have small impacts for our studies of cortical circuits using the $Nr5a1^{Cre}$ line to define starter cells. The first of these, shared common inputs, is small in cortex due to the sparse connectivity and the sparsity of starter cells in our experiments, but at most might cause SF to be about 20% more than IF (see Materials and methods for detailed consideration). The second factor is more complex, but our analyses show that for cortical circuits it is simplified by the fact that the great majority of connections are mediated by a single synaptic contact (*Figure 5*). As detailed in Materials and methods along with *Figures 5–6*, overall IF is likely to be at least 78% of SF (ratio of IF/SF = 0.78). With IF/SF = 0.78 our measured SF of 0.4 corresponds to 31% of input neurons labeled (IF = 0.31) and the probability of rabies virus spread across a single synaptic contact is estimated as 0.28 (U=0.28).

## Discussion

Genetically modified RVdG has proven instrumental in deciphering the intricate connectivity patterns of neural networks. Here, we investigated the efficiency with which RVdG spreads retrogradely across excitatory synapses and report, for the first time, direct measurements of the proportion of starter neurons' postsynaptic densities whose presynaptic inputs are labeled (SF) using the monosynaptic

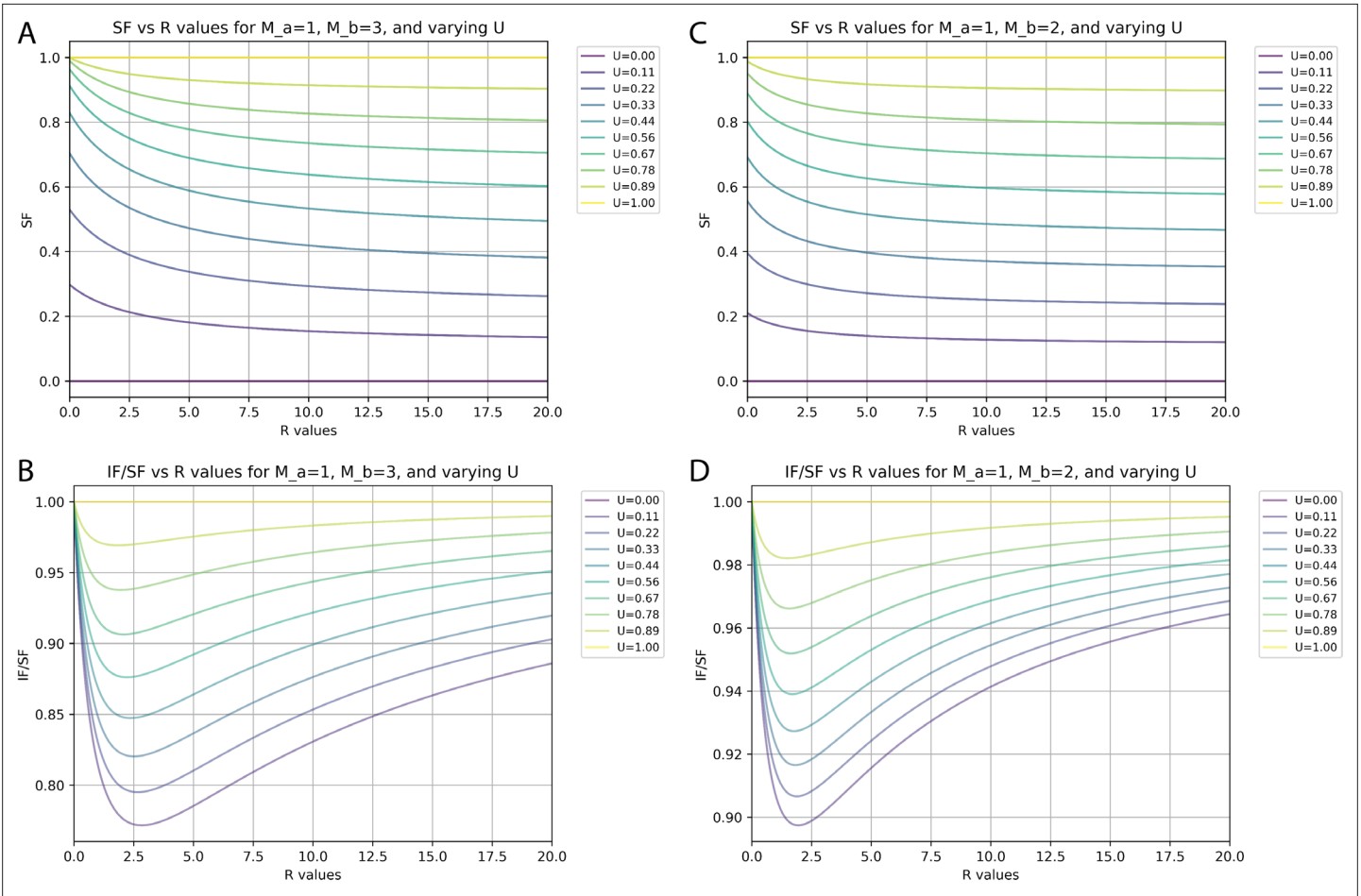

**Figure 6.** Relationships between synaptic fraction (SF), input fraction (IF), and unitary synaptic efficiency (U) depend on synapses per input neuron (M) and proportions of neurons with differences in M. (**A** and **C**) SF against the ratio (R) of neurons with M values of 1 versus either 3 (**A**) or 2 (**C**) for different values of U. (**B** and **D**) Ratio of IF versus SF against the R of neurons with M values of 1 versus either 3 (**B**) or 2 (**D**) for different values of U.

rabies tracing system. Our results demonstrate that with the particular AAV helper viruses, AAV titers, and EnvA+ RVdG titers and strain we used, about 40% of excitatory synaptic contacts onto the examined cell types were labeled. Further considerations detailed in the Results and Materials and methods indicate that the measured SF of 40% likely corresponds to at least 30% of input neurons labeled (IF = 0.30) and a probability of rabies spread across individual synapses (unitary efficiency, U) of 28%.

Of note, we found that efficiency of spread did not vary depending on the distance of synapses to the cell body or across dendrite types. We were concerned that more distal synapses might be less likely to be labeled than proximal synapses because antibody staining for rabies particles shows that during the first 1–3 days post-infection, particles are concentrated around the cell body; but they are not detected more distally until later time points (***Nassi and Callaway, 2006***). In the experiments described here, efficiency of spread was quantified at 7 days post-infection. It therefore appears that any initial bias for spread at proximal synapses might be eliminated by use of longer survival times. Since we only compared efficiency at one typical time point, care should be taken in interpreting results at shorter times post-infection. The same factors that we have considered for their impact on the relationships between measured SF, IF, and U are also important to consider with respect to the spatial locations of labeled synapses. This is because synaptic labeling resulting from either shared inputs or multiple synaptic contacts from a single input neuron are not the result of the spread of rabies across a synaptic contact. Our analyses show that the proportions of labeled synapses arising from these mechanisms is small relative to the variability and proportions of observed SF at distal versus proximal synapses (***Figures 3G and 4F***).

Previous studies quantifying the numbers of rabies-labeled input cells per starter cell (CI) have identified several different factors that can increase or decrease the efficiency of transsynaptic spread. While there are few direct comparisons with matched titers of rabies viruses and helper viruses, both the strain of rabies virus and the strain and variant of rabies glycoprotein are likely to be influential. Importantly, when a particular rabies strain and glycoprotein are used, the CI appears to depend most strongly on the levels of G expressed in the starter cells (*Callaway and Luo, 2015*). Studies suggest that maximizing G expression, either through the use of strong promoters (*Miyamichi et al., 2013*) or optimized helper virus concentration (*Lavin et al., 2020*), increases spread efficiency. Importantly, the use of 2A linker elements has been implicated in reduced expression of G (*Wall et al., 2010*; *Watabe-Uchida et al., 2012*). Thus, the use of separate AAVs, with one expressing glycoprotein independently from other genes, is recommended to improve efficiency.

A second factor likely to influence efficiency is the number of rabies particles entering a starter cell (*Callaway and Luo, 2015*). The variability between individual cells that we sampled (*Figures 2G and 3C*) is likely to reflect these differences. For example, there is expected variability in the numbers of oG-expressing AAV copies and EnvA+ RVdG particles entering each starter cell. Despite the variability that we observe between individual cells, under matched conditions these differences converge to similar means when averaged across even modest numbers of cells (typically 9 cells in our sample) that are far lower than the number of total starter cells in a typical animal. It is noteworthy that it would be straightforward to increase the SF well above the 35–40% we observed by simply increasing the levels of oG expression. For example, our AAV helper viruses were designed to allow unambiguous detection of starter cells for accurate quantification, and we therefore utilized a 2A linker element to express both nuclear BFP and oG from a single helper virus. For experiments where unambiguous starter cell detection is not crucial, oG expression can be increased by expression as a single protein from an alternate helper virus. Additionally, oG expression can be further increased through use of an expression amplification system such as TRE/tTA (*Lavin et al., 2020*).

These observations suggest that the efficiency of rabies tracing can be manipulated over a wide range depending on experimental goals. But counterintuitively, the ideal rate of spread is probably far less than 100% for most experiments. This is because the spread of rabies is likely to depend on the numbers of synaptic contacts as well as the size of the postsynaptic density, both of which are related to functional connection strength (*Holler et al., 2021*; *Murthy et al., 2001*). Thus, if a reasonably small fraction of inputs were labeled, the numbers of labeled presynaptic cells of different types would be expected to scale with the strength of connections from each cell type. But if rabies were to spread across 100% of synaptic contacts, this sensitivity would be lost; every input cell would be labeled regardless of connection strength. On the other hand, if efficiency were exceedingly low one would expect more variability between animals and the potential to fail to detect inputs from weak connections. As considered in our formal modeling of the relationships between proportion of synaptic contacts labeled, proportion of inputs labeled, and efficiency of rabies spread across individual synaptic contacts, a value of 40% of contacts labeled (SF = 0.4) likely corresponds to about 30% of inputs labeled (IF = 0.30) and a unitary synaptic efficiency of about 28% (U=0.28). We therefore suggest that values in the range of 35–50% of synaptic contacts labeled are likely to be near optimal in providing both large numbers of labeled input neurons and some degree of sensitivity to connection strength (*Henrich et al., 2020*). The equations that we derived (see Materials and methods) can also be used to provide insight into the relationships between numbers of synaptic contacts per input neuron and the relative probability that it will be labeled in comparison to other neurons with a different number of synaptic contacts. These equations can be used to assess circuit configurations that are likely to be different in other brain areas than for the cortical circuits on which we have focused.

Considering the finding that rabies labels less than half of synaptic inputs, it is likely that differential spread of rabies virus leads to differences in the probability of distinct input types being labeled. Although we found evidence that the subcellular location of synaptic contacts of input cells onto the dendrites of starter cells does not affect spread efficiency, it remains unknown what leads to the observed incomplete labeling. It is possible that this is affected by the number of synaptic contacts a presynaptic neuron makes onto the starter cell, differences in uptake receptors on the axon of distinct input cell types, or differences in postsynaptic density size (*Callaway and Luo, 2015*). Indeed, one limitation of our experimental design is that our light microscopy images do not allow accurate

measurement of postsynaptic density size, which would require electron microscopy. Furthermore, it is important to note that we only examined spread efficiency at excitatory synapses labeled with the excitatory postsynaptic density marker PSD-95 and did not quantify the efficiency of spread to inhibitory presynaptic inputs. It should also be noted that the locations and numbers of postsynaptic sites were assessed based on expression of PSD-95-GFP from the rabies virus. If this overexpression were to somehow result in new puncta that do not have a presynaptic input this could result in an underestimate of proportion of actual PSDs labeled.

Overall, we report the efficiency of spread achieved in multiple conditions consisting of distinct starter neuron types. Additionally, the described approach can be used by other researchers to test efficiency of spread across different rabies reagents, distinct input cell types, and starter neuron types not tested in this study.

# Materials and methods

## Mouse transgenic lines

All experimental procedures were approved by the Salk Institute Animal Care and Use Committee (IACUC). All the animals were handled according to the approved IACUC protocol 13-00005. C57BL/6J mice were used as wild-type. GENSAT BAC transgenic Nr5a1-Cre (Jackson Laboratory stock # 006364), knock-in Som-IRES-Cre (Jackson Laboratory stock #013044), knock-in Vip-IRES-Cre (Jackson Laboratory stock #028578), and GENSAT BAC transgenic Sim1_KJ18-cre mice have been previously described (*Gerfen et al., 2013*; *Harris et al., 2014*; *Taniguchi et al., 2011*). Transgenic mice were maintained on C57BL/6J backgrounds. Mice were housed with a 12 hr light and 12 hr dark cycle and ad libitum access to food and water. Both male and female mice were used for experiments.

## Virus preparation

The following AAVs were produced by the Salk GT3 Viral Core: AAV8-hSyn-FLEX-H2BmTagBFP2-oG (3.57E+13 GC/ml) and AAV8-nef-AO-66/71-TVA950 (5.25E+13 GC/mL). AAV1-CAG.FLEX.GFPsm_myc.WPRE.SV40 (1.12E+13 GC/ml) was purchased from Addgene. EnvA+ RVdG-5PSD95eGFP-SynPhRFP (1.55E+08 IU/ml) was produced by the Salk GT3 Viral Core. AAVs and rabies virus can be purchased from the Salk Viral Vector Core.

## Animal surgery for virus injection

For rabies transsynaptic spread efficiency experiments, mice received AAV helper injections at postnatal day (P) 50. Mice were initially anesthetized with 2% isoflurane and maintained at 1.5% isoflurane after placement on a stereotax (David Kopf Instruments, Model 940 series) for surgery and stereotaxic injections. A small craniotomy was made with a mounted drill over the primary visual cortex of the left hemisphere using the following coordinates: 3.4 mm posterior and 2.6 mm lateral relative to bregma. For transsynaptic tracing experiments AAV8-hSyn-FLEX-H2BmTagBFP2-oG, AAV8-nef-AO-66/71-TVA950, and AAV1-CAG.FLEX.GFPsm_myc.WPRE.SV40 were mixed at a ratio of 1:40, 1:100, and 1:2 to final concentrations of 8.92E+11, 5.25E+11, and 5.6E+12 GC/ml respectively. For control experiments with no oG AAV8-nef-AO-66/71-TVA950 and AAV1-CAG.GFPsm_myc.WPRE.SV40 were mixed to a titer matched concentration of 5.25E+11 and 5.6E+12 GC/ml. 100 nl of mixture was injected into the center of V1 0.5–0.7 mm ventral from the pia using a pulled glass pipette with a tip size of 30 µm connected to a 1 ml syringe with 18 G tubing adapter and tubing. To prevent backflow, the pipette was left in the brain for 5 min after injection. Three weeks after AAV helper injection, 150 nl of EnvA+ RVdG-5PSD95eGFP-SynPhRFP was injected into the same site in V1. After recovery, mice were given water with ibuprofen (30 mg/kg) and housed for 7 days to allow for transsynaptic rabies spread and fluorescent protein expression.

## Histology

Seven days after rabies injection, brains were harvested after transcardial perfusion using PBS followed by 4% paraformaldehyde (PFA). Brains were dissected out from skulls and post-fixed with 2% PFA and 15% sucrose in PBS at 4°C for 16–20 hr, then immersed in 30% sucrose in PBS at 4°C before sectioning. 50 µm coronal brain sections were prepared using a freezing microtome. Free-floating sections were incubated at 4°C for 16 hr with rabbit anti-Myc (1:500, C3956; Sigma-Aldrich) primary

antibody in PBS/0.5% normal donkey serum/0.1% Triton X-100, followed by donkey anti-rabbit conjugated to Alexa 647 (1:500, A-21206, Thermo Fisher) at room temperature for 2–3 hr. Immunostained tissue sections were mounted on slides with polyvinyl alcohol mounting medium containing DABCO and allowed to air-dry overnight.

For characterization of the population of Cre-expressing neurons in *Nr5a1*[Cre] mice, these animals were crossed with AI14 reporter mice (Jackson Laboratory stock #007914) to yield TdTomato expression in all Cre-expressing neurons (***Wang et al., 2023***). Three adult animals (73–193 days of age) were perfused and brains removed, and sectioned as described above. Sections were processed as follows: antigen retrieval was performed using 10 nM sodium citrate buffer and then sections were immunostained with rabbit anti-NeuN (1:1000, ab177487, Abcam), mouse anti-GAD67 (1:1000, MAB5406, MilliporeSigma), rat anti-RFP (1:1000, 5F8-100, Chromotek) followed by Alexa 488 donkey anti-rabbit (1:1000, A21207, Thermo Fisher), Alexa 647 donkey anti-mouse (1:500, A31571, Thermo Fisher), and CF568 donkey anti-rat (1:500, 89138-546, Biotium), secondaries. Sections were first imaged on an Olympus BX63 microscope using a 10×/0.4 NA objective (Olympus) to identify the borders of V1 and layer 4. Sections for cell counting were then imaged on a Zeiss LSM880 confocal microscope using a 20×/0.8 NA objective. A stitched z-stack image was taken so that the entirety of V1 in one of the hemispheres of that section could be used for quantification. Within these regions the x,y,z coordinates of all RFP-positive (TdTomato) cells were marked and then these coordinates were used to calculate nearest-neighbor distances and the numbers of labeled neurons located with 50 µm of each labeled neuron. Across 774 labeled neurons whose positions were marked in 7 sections from 3 animals, we find that both the mean and median numbers of labeled neurons located within 50 µm of any given labeled neuron of interest is 7. Additional analyses also confirmed that only GAD-negative excitatory neurons were labeled.

## Image acquisition and analysis

Individual sections were first scanned at ×10 magnification using an Olympus BX63 widefield fluorescent microscope to detect starter neurons triple positive for H2BBFP-oG, smFP_myc, and RVdG-PSD95GFP-SynPhRFP. Individual starter neurons were then imaged using a Zeiss LSM 880 Airyscan FAST Microscope with a Plan-Apochromat 63×/1.4 Oil DIC M27 objective. First, neurons were confirmed to be starter neurons by checking for expression of nuclear BFP. Z-stacks of images were acquired with a step interval of 500 nm for select dendritic domains. Dendrites immediately next to the soma and spanning 75 µm away were defined as proximal dendrites and representative images were acquired. Using expression of smFP_myc, dendrites were traced from the soma up to the pial surface. Dendritic regions spanning 75 µm from the pial surface were acquired and designated as distal domains. Images were processed and analyzed using NIH ImageJ software (FIJI). Airyscan super-resolution imaging can resolve structures ~120 nm apart in the lateral dimension and ~350 nm apart in the axial dimension. The synaptic cleft size, or space between the pre- and postsynaptic density, ranges from 20 to 30 nm, making it outside of the range of the resolution limit. Therefore, we required partial overlap, or colocalization, between pre- and postsynaptic markers and did not allow for any distance between the two markers to consider them a synapse. For synaptic counting, images were first analyzed with only the far-red (smFP_myc) and green channel visible to count all PSD95GFP-labeled postsynaptic densities colocalized with the far-red labeled dendritic region. After postsynaptic counts for that region were quantified, the red channel was turned on to quantify the number of presynaptic densities colocalized with postsynaptic densities. Due to the abundance of synapses across a 50 µm coronal section, puncta were counted manually one z-plane at a time. Images presented in manuscript are max intensity projection reconstructions to allow for 2D visualization of 3D structures. Wilcoxon rank-sum test for non-parametric comparisons was used for statistical analysis. For all figures: ***, $p < 0.001$; ns = not significant, $p > 0.05$.

## EM analysis methods

Analysis of synaptic connectivity statistics from structural electron microscopy was done on the publicly available dataset of mouse visual cortex from the MicRonS project (***Bae et al., 2023***). To assess the number of synapses per excitatory connection, neurons were surveyed from the collection of cells that were annotated as having 'clean' or 'extended' axons, which reflect the set of axons which have been removed of false mergers. We included only neurons which had more than 100 synaptic

outputs and were excitatory. For each excitatory cell, between 65% and 90% of its synapses could be mapped to a post-synaptic soma (*Bae et al., 2023*). Most post-synaptic somas have been cell typed in the dataset based on a model trained on soma and nucleus features (*Elabbady et al., 2022*), using labels provided by a combination of human labeling and unsupervised clustering (*Schneider-Mizell et al., 2023*). Therefore, we were able to calculate the average number of synapses per excitatory connection as a function of cell type for many cell-type interactions.

Further, for assessing shared connections onto layer 4 cells specifically, we calculated the distance from each post-synaptic soma to others layer 4 somas in the volume, and then measured which of those had connections from the same pre-synaptic axon. We then calculated the mean connected fraction as a function of distance between the two post-synaptic somas. All data analysis was done in python using the CAVEclient to access the dataset (*Dorkenwald et al., 2023*) and numpy (*Harris et al., 2020*), pandas (*McKinney, 2010*; *team, T. pandas development, 2023*), and scipy (*Virtanen et al., 2020*) to calculate these values.

## Calculations of SF and IF

We conducted formal analyses of how several different factors in the organization of cortical circuits might influence the relationship between the proportion of postsynaptic specializations on a given starter neuron that have rabies-labeled presynaptic terminals (SF), the probability that rabies spreads across a single synaptic contact (U), and the proportion of input neurons to the starter neuron that are labeled (IF). We consider two different influences that are both expected to result in measured values of SF that are larger than IF. These are: (1) the extent to which different starter neurons share common inputs; and (2) the distribution of the numbers of synaptic contacts provided per input neuron onto a particular starter neuron.

### The potential impact of common inputs

Shared common inputs are expected to result in an SF value that is larger than the input proportion labeled (SF>IF). This is because the divergent connections of an input neuron can result in the labeling of a synaptic contact on a starter neuron even if rabies virus never spreads from the starter neuron at that particular synaptic contact. Specifically, rabies can spread from a different starter neuron resulting in expression of synRFP in the divergent input neuron. That synRFP could then result in labeling of a synaptic contact without any spread of rabies virus across the labeled contact. The rate of synaptic labeling on a given starter cell by this indirect route depends on three factors: the numbers and spatial distribution of other starter cells; the proportion of common inputs that are shared between starter cells at different distances; and the probability that inputs to a starter cell are labeled (IF).

We directly assessed the maximum possible numbers of other layer 4 starter cells and their spatial distributions by analyzing the density and distribution of Cre-expressing neurons in *Nr5a1*^Cre mice. And we analyzed a 3D reconstructed EM volume to directly measure the probability that a neuron presynaptic to a layer 4 excitatory neuron also provides a common input to a neighboring layer 4 neuron (*Figure 5A*).

To calculate the proportion of the measured value of SF that arises from common input we sum together the contributions of all neurons providing shared input and then multiply by the estimated proportion of those neurons that were labeled (IF). Using an estimated IF of 0.3 (IF = 0.75 SF from common input and multiple contacts combined, see below), we obtain the following proportions of the measured value of SF that arise from common inputs at different distances from the subject neuron. At 0–50 µm the proportion of shared input is 5% (*Figure 5A*) and there are 7, nr5a1 neurons. If the rate at which inputs are labeled (IF) is 0.3 (see above and below) then 30% of the 5% shared inputs = 1.5% more labeled synapses can potentially be added indirectly by a second nearby starter neuron. But if 30% of the neurons providing input to the cell of interest were already labeled directly, then only 70% of the potential additions, or 1%, will be new additions. As additional nearby starter cells are incorporated into the network and their inputs become labeled, each new cell will add a progressively smaller percentage of new indirect inputs because the portion of previously unlabeled synapses will decrease; but for small numbers of input neurons, 1% remains a close approximation. For 7 other starter cells located within 50 µm of the subject neuron the total maximum contribution (assuming all nr5a1 neurons were rabies-infected starters) would be 7% of the measured SF value. At 50–100 µm there are 13 additional nr5a1 neurons and the rate of shared input is 3.3%, yielding a total

contribution of 9%. At 100–200 μm there are an additional 28 potential starter neurons and the rate of shared input is 1%, yielding a total contribution of 6%. Contributions beyond 200 μm are negligible (less than 1%) due to the very low rate of common input (~0.1%). Altogether, the maximum possible proportion of labeled synapses that result from shared input is about 0.22 (0.07+0.09+0.06). Thus, this factor would be expected to result in SF being at most 22% greater than IF (IF = 0.78 SF). Since the proportion of nr5a1 neurons that are directly infected with rabies and serve as starters is almost certainly less than 1 and probably closer to 0.5, this is an overestimate. If only half of the nr5a1 neurons are starter cells then the proportion of labeled synapses resulting from shared input would be only 0.11 and IF/SF = 0.89.

## The impact of the distribution of the numbers of synaptic contacts provided per input neuron onto a particular starter neuron

We next consider how the numbers of synaptic contacts provided by each input neuron onto a given starter neuron will influence the relationship between SF and IF. Overall our consideration of this issue indicates that this will have only a small effect on our measures of excitatory inputs to cortical neurons because more than 90% of such connections involve only a single synaptic contact (see below). Regardless of the actual configuration of inputs, the maximum value of IF will be SF (SF = IF), but for most configurations SF is larger than IF (SF>IF; details below). It is informative to first consider scenarios in which SF = IF because this is not entirely intuitive. The most intuitive case is a configuration in which each input neuron invariably provides just one synaptic contact to the starter neuron. Here, it is straightforward to infer that SF and IF are equal because the only way to label an input neuron is for rabies to spread from the single synaptic contact and then for SynRFP to spread to and label that same synaptic contact. But for cases in which some or all input neurons provide more than one synaptic contact onto a starter neuron, the possible scenarios are less intuitive. Generally, such a configuration will cause IF to be less than SF (but not always). For example, rabies virus might spread from one of the contacts provided by an input neuron and this would then result in production of synRFP that labels all of the contacts provided by that input neuron. Taken at face value, one would expect that this would cause SF to be larger than IF; this is usually, but not always the case. In an extreme scenario, suppose that each input neuron has multiple synaptic contacts, but the number of synaptic contacts is equal for each neuron. In this case, SF will be equal to IF. For example, suppose a starter neuron has a total of 1000 synaptic contacts (S) that arise from 200 input neurons (N) each providing exactly 5 contacts. Even if rabies virus spreads from just 1 of the 5 contacts provided by an input neuron and then labels all 5 contacts, the proportion of postsynaptic specializations labeled (SF) will always be equal to the proportion of input neurons labeled (IF), regardless of how efficiently rabies virus spreads between neurons. If N*=number of labeled input neurons and S*=number of labeled synaptic contacts, then IF = N*/200; SF = S*/1000; S*/N*=5; therefore SF = IF. This relationship holds for all cases of S/N provided that all input neurons have the same S/N.

The actual configuration of cortical circuits is likely more complex and the precise features are not known for all input sources. However, available data based on a reconstructed EM volume (see EM analysis methods) indicate that in adult mouse visual cortex, it is rare for a local connection from an excitatory neuron onto another cortical neuron to involve more than one synaptic contact. *Figure 5B* plots the distribution of the numbers of synaptic contacts per connection for 76,678 excitatory neuron to excitatory neuron connections, involving 503 pre-synaptic cells and 26,079 post-synaptic cells (note the log scale of the Y-axis in *Figure 5B*). Out of 76,678 connections, only about 6000 (~8%) involve more than one synapse and only about 1000 (~1.3%) involve 3 or more synapses. The average number of synapses per connection is 1.0965 (std = 0.3726). Note that these values are much lower than observed in previously available data based on connections that were first identified by electrophysiology in brain slices from 12- to 23-day-old rats, followed by EM reconstruction of the labeled neurons (*Feldmeyer et al., 1999*; *Feldmeyer et al., 2002*; *Feldmeyer et al., 2006*). We consider the newer, much more extensive, and relatively unbiased data from adult mouse visual cortex to be more relevant to our experiments in adult mouse visual cortex. Nevertheless, we consider the potential impact of larger values since the equations we derive and the examples that we illustrate can be informative for future consideration of labeling in other circuits and cell types.

Here, we describe how the distribution of input ratios (S/N=synapses per input neuron) influences the relationship between SF and IF. We refer to S/N as multiplicity, or M. For simplicity, we consider

only cases with two different populations of input neurons (a and b), each with different numbers of synapses per neuron ($M_a$ and $M_b$). With this scenario the relationship between SF and IF depends on both the proportion of input neurons in the a and b populations (ratio, $R=N_a/N_b$) and on the probability that rabies virus spreads across a single synaptic contact (unitary synaptic efficiency, U). Below, we provide equations and their derivation that allow for calculation of both SF and IF given: (1) the ratio ($R=N_a/N_b$) of the numbers of neurons (N) in population a ($N_a$) and in population b ($N_b$); (2) the multiplicity of synaptic contacts per neuron (M) for population a ($M_a$) and population b ($M_b$); and (3) the unitary synaptic efficiency (U).

At an extreme, there might be just 1 S/N for one population (a, $M_a = 1$) and 3 S/N for another population (b, $M_b = 3$). *Figure 6A and B* plots the calculated values of SF and IF/SF, respectively, for different ratios (R) of $N_a/N_b$ and for different levels of unitary synaptic efficiency (U) when $M_a = 1$ and $M_b = 3$. Under these conditions and with a worst-case estimate of U=0.22 (our measured SF of 0.4 suggests actual U is closer to 0.28, see below), the smallest possible IF/SF value is ~0.8. If these were the actual values, then our measured SF of 0.4 would correspond to an IF value of 0.32, or 32% of all inputs labeled.

We consider a scenario with $M_a = 1$ and $M_b = 2$ to provide a closer simulation of the actual configuration of cortical circuits since values of 1 and 2 synapses per neuron account for nearly 99% of all connections (*Figure 5B*). *Figure 6C and D* plots the calculated values of SF and IF/SF, respectively, for different ratios (R) of $N_a/N_b$ and for different levels of unitary synaptic efficiency (U) when $M_a = 1$ and $M_b = 2$. Here, we see that the smallest IF/SF is now calculated when R is about 2 (*Figure 6B*) and that with R=2.0 our measured SF values of around 0.4 are typically calculated when U~=0.28 (*Figure 6A*). With U=0.28 and R=2.0, IF/SF in this scenario is about 0.91 (*Figure 6B*). Finally, if we consider that based on the actual measurements for cortical circuits the value for R is at least 10 (*Figure 5B*), our measured SF value of 0.4 corresponds to U~=0.35 (*Figure 6C*). With U=0.35 and R=10, IF/SF is about 0.95 (*Figure 6D*). Therefore, under this 'most plausible' scenario considering the impact of multiple synaptic contacts, our measured SF of 0.4 suggests that about 38% of input neurons are labeled (IF = 0.38) and that the probability of rabies virus spread across a single synaptic contact is 35% (U=0.35).

These estimates of IF and U are made independently from the effects of common input, which, as detailed above, are expected to result in IF being at most 78% of the measured value for SF. If we incorporate an additional maximum impact due to multiple synaptic contacts based on IF/SF = 0.95, then we obtain an overall estimate of 0.78×0.95 = 0.74=IF/SF. With measured values of SF = 0.4 we estimate IF ~=0.3. Calculations above indicate that the impact of multiple synapses per contact will result in IF that is about 8% larger than U. Therefore, we estimate that, at a minimum, U=0.28.

## Derivation of equations relating SF to IF

We considered conditions in which there are two different populations of neurons, a and b, that each have different numbers of synaptic contacts (S) onto the starter neuron per input neuron (synapses per neuron, S/N, or multiplicity M). We assume that when there are multiple synaptic contacts from an input neuron to a starter neuron, the probability that rabies virus spreads across a synaptic contact (U) is the same for all contacts from both input populations and is independent for each contact. Further, if rabies spreads across one of an input neuron's contacts, the synRFP expressed from rabies virus in that neuron spreads to and labels all the synaptic terminals from that input neuron.

As shown in the equations and derivation below, the values for SF and IF depend on: (1) the ratio (R) of the numbers of neurons (N) in population a ($N_a$) and population b ($N_b$); (2) the numbers of synaptic contacts per input neuron (multiplicity, M) for population a ($M_a$) and population b ($M_b$), and the probability that rabies virus will spread across an individual synaptic contact (U).

> N=number of presynaptic neurons; $N_{tot} = N_a + N_b$
> S=number of synaptic contacts on the postsynaptic neuron; $S_{tot} = S_a + S_b$
> R=ratio of the numbers of presynaptic neurons; $R = \frac{N_a}{N_b}$ ; $N_a = R * N_b$
> M=number of synaptic contacts made by each presynaptic neuron; $M_a$ or $M_b$
> U=unitary synaptic efficiency, probability of rabies spread across a single synaptic contact
> p=probability of labeling a presynaptic neuron; $p_a$ or $p_b$
> N*=number of labeled presynaptic neurons; $N_a^*$ or $N_b^*$
> S*=number of labeled synaptic contacts; $S_a^*$ or $S_b^*$

$$p_x = 1 - (1 - U)^{M_x}$$

$$N_x^* = N_x * p_x$$

$$N_a^* = R * N_b * p_a$$

$$S_x^* = N_x^* * M_x$$

$$IF = \frac{N_{tot}^*}{N_{tot}} = \frac{N_a^* + N_b^*}{N_a + N_b} = \frac{(N_a * p_a) + (N_b * p_b)}{N_a + N_b} = \frac{(N_b * R * p_a) + (N_b * p_b)}{(N_b * R) + N_b} = \frac{(R * p_a) + p_b}{R + 1}$$

$$S_{tot}^* = S_a^* + S_b^* = (N_a^* * M_a) + (N_b^* * M_b) = (N_b * R * p_a * M_a) + (N_b * p_b * M_b) = N_b * [(R * p_a * M_a) + (p_b * M_b)]$$

$$S_{tot} = S_a + S_b = (N_a * M_a) + (N_b + M_b) = (R * N_b * M_a) + (N_b + M_b) = N_b * [(M_a * R) + M_b]$$

$$SF = \frac{S_{tot}^*}{S_{tot}} = \frac{N_b * [(R * p_a * M_a) + (p_b * M_b)]}{N_b * [(M_a * R) + M_b]} = \frac{(R * p_a * M_a) + (p_b * M_b)}{(M_a * R) + M_b}$$

## Statistics and reproducibility

No statistical methods were used to pre-determine sample sizes; rather sample sizes were established based upon similar studies in the literature. Stereotaxic injections were repeated in three biological replicates for each mouse line, and both male and female mice were used. All statistical tests were performed using R version 4.1.1 and details of individual tests are described in figure legends. All data are reported as the mean ± SEM, except synapses per connections which is mean ± std.

## Acknowledgements

We thank all Callaway Lab members for discussion. We thank Helen Wang and Oyshi Dey for generating and sharing data related to Nr5a1 cell counts. We also thank the Salk Viral Vector and Biophotonics Core staff members. This work was supported by NSF grant DBI-1707261 and NIH Diversity Supplement to R01 EY022577 (EMC). MP was supported by the Paul and Daisy Soros Fellowship for New Americans and NIH T32 GM007198. Forrest Collman acknowledges support from NSF NeuroNex 2 award 2014862 and the National Institute of Neurological Disorders and Stroke of the National Institutes of Health under Award Number U24NS120053.

## Additional information

### Funding

| Funder | Grant reference number | Author |
|---|---|---|
| National Science Foundation | DBI-1707261 | Edward M Callaway |
| National Eye Institute | R01 EY022577 | Edward M Callaway |
| Paul and Daisy Soros Fellowships for New Americans | | Maribel Patiño |
| National Institute of General Medical Sciences | T32 GM007198 | Maribel Patiño |
| National Science Foundation | 2014862 | Forrest Collman |
| National Institute of Neurological Disorders and Stroke | U24NS120053 | Forrest Collman |

The funders had no role in study design, data collection and interpretation, or the decision to submit the work for publication.

## Author contributions
Maribel Patiño, Conceptualization, Formal analysis, Supervision, Funding acquisition, Investigation, Methodology, Writing – original draft, Writing – review and editing; Willian N Lagos, Neelakshi S Patne, Paula A Miyazaki, Investigation, Writing – review and editing; Sai Krishna Bhamidipati, Formal analysis, Writing – review and editing; Forrest Collman, Resources, Formal analysis, Investigation, Visualization, Writing – review and editing; Edward M Callaway, Conceptualization, Resources, Formal analysis, Funding acquisition, Methodology, Project administration, Writing – review and editing

## Author ORCIDs
Maribel Patiño ⬤ http://orcid.org/0000-0003-0820-3315
Sai Krishna Bhamidipati ⬤ http://orcid.org/0000-0003-3498-3730
Forrest Collman ⬤ https://orcid.org/0000-0002-0280-7022
Edward M Callaway ⬤ https://orcid.org/0000-0002-6366-5267

## Ethics
All work involving live animals was approved by the Salk Institute Animal Care and Use Committee (IACUC). All the animals were handled according to the approved IACUC protocol 13-00005.

## Decision letter and Author response
Decision letter https://doi.org/10.7554/eLife.89297.sa1
Author response https://doi.org/10.7554/eLife.89297.sa2

## Additional files

### Supplementary files
• MDAR checklist

• Source data 1. Raw data of pre-and post-synaptic densities for *Figures 2–4*. Raw data includes counts for Nr5a1, Som, and vasoactive intestinal peptide (Vip) starter cells and control experiments omitting oG.

### Data availability
All data generated or analyzed in this study are included in figures and Source data 1.

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
