## [Editor Report]

This study provides valuable new information on using monosynaptic rabies virus for trans-synaptic tracing, a widely used tool to map synaptic inputs to specific types of neurons in the mammalian brain. The authors provide convincing quantitative information about the efficiency for the rabies virus to cross synapses of visual cortical neurons in the mouse. They further demonstrate that the cortical cell type and location of synapse in the postsynaptic cell do not appear to affect the efficiency for the rabies virus to cross synapses.

---

## [Decision Letter]

**Decision letter after peer review:**

[Editors’ note: the authors submitted for reconsideration following the decision after peer review. What follows is the decision letter after the first round of review.]

Thank you for submitting the paper "Quantification of Monosynaptic Rabies Tracing Efficiency" for consideration by *eLife*. Your article has been reviewed by 3 peer reviewers, one of whom is a member of our Board of Reviewing Editors, and the evaluation has been overseen a Senior Editor. The reviewers have opted to remain anonymous.

We are sorry to say that, after consultation with the reviewers, we have decided that this work will not be considered further for publication by *eLife*.

Comments from all three reviewers are attached. The reviewers also had a robust discussion session after all reviews were submitted. While all three reviewers appreciate the value of your work, the consensus is that the advance is not sufficient for a full research article. In particular, the key measurement of "labeling efficiency at synaptic level" is very easily confused by readers as "the efficiency with which rabies virus crosses synapses", which is a more valuable measure that is less dependent on specific connectivity architectures of specific circuits.

We regret that we cannot consider the current manuscript further. However, if you are able to thoroughly address reviewers' critiques with further work, you are welcome to submit a revised manuscript to *eLife* as a new submission. We will try our best to recruit the same reviewers.

*Reviewer #1 (Recommendations for the authors):*

G-deleted rabies virus has been widely used to trace monosynaptic inputs to specific populations of neurons in the mammalian brain since its first report by Callaway and colleagues in 2007. Despite its wide use, many fundamental properties of the method remain to be characterized. In this short report, Patino et al. addressed two important questions: (1) what fraction of synapses does the rabies virus cross from a starter cell? (2) does this vary for synapses located at different distances to the soma? Answers to these questions should help researchers to interpret their data and thus will be of general interest to the neural circuit community.

To answer these questions, the authors produced a new rabies virus that expresses both a presynaptic marker tagged with RFP and a postsynaptic marker (for excitatory neurons) tagged with GFP, and simply counted within starter cells the fraction of GFP+ puncta that have an adjacent RFP punctum. Since most RFP puncta are a result of the trans-synaptic transfer of rabies virus (the authors did a nice control of skipping G in one experiment), the authors can "quantify efficiency of spread at the synaptic level". The authors found about 35-40% of GFP+ puncta have RFP puncta next to them, and interestingly this number does not change whether these quantifications were performed close to the cell body or far from the cell body (thus answering question #2). However, the current version has a number of issues that should be addressed.

1. While the authors conclude properly that under their experimental conditions, "35% to 40% of excitatory inputs to the examined cell types were labeled," many readers would mistakenly think that "rabies virus crosses 35-40% of synapses of the starter cells" from reading this paper. The latter is unlikely true for two reasons. First, if starter cell 1 transfers rabies virus to presynaptic neuron 1, which not only connects with starter cell 1 but also starter cell 2, then a pair of green/red puncta between starter cell 2 and presynaptic neuron 1 could result even if rabies virus never crossed the synapse between starter cell 2 and presynaptic neuron 1. Second, if starter cell 1 forms 5 synapses with presynaptic neuron 1, then it is possible that the rabies virus just crossed 1 of the 5 synapses (there may be some data supporting 5-10 synapses between synaptic pairs of cortical excitatory neurons), and the other 4 pairs of puncta are incidental. Thus, the actual fraction of synapses that the rabies virus crosses is likely substantially below 35-40%. While the first caveat can be overcome by doing experiments from single starter cells, the second caveat cannot be avoided. It is possible that the authors could estimate -based on cortical connectivity data (luckily V1 is one of the best studied in this regard, including serial EM reconstruction)-"what fraction of synapses does the rabies virus cross," which is a more fundamental question likely generalizable across cell types than "what fraction of excitatory inputs are labeled," which will vary greatly across circuits because of specific connection properties. At the very least, the authors should distinguish these two questions clearly, and state why they can differ, so that readers won't come away being misled.

2. I am confused by Figure 2F, one of the most important panels of the entire paper that provides examples of GFP/RFP co-staining at high resolution. GFP marks postsynaptic density and thus should colocalize with the dendritic marker (magenta) and RFP marks presynaptic terminals of presynaptic neurons and should be adjacent but outside the magenta-positive area. But the third row (the positive example) shows the exact opposite. Red is within the magenta area, whereas green is adjacent to the magenta area. Can the authors clarify?

3. This brings an important issue of setting objective criteria for synaptic pairs: how much overlap, what's the maximum/minimum distance, etc, as this is the basis for the entire study. Indeed, what counts as a synaptic pair based on pre and postsynaptic markers in light microscopy is a debated issue. The authors are helped here by the sparse labeling of starter cells and presynaptic partners; still, I highly recommend that the authors state clearly the criteria in Methods.

4. The negative control of omitting oG in Figure 2 is very nice; I hope that the authors will do the same for tracing inputs to the two types of inhibitory neurons. These inhibitory neurons are also known to form gap junctions and it will be interesting to see whether the rabies virus would jump through gap junctions without oG, even if they are not known to form substantial chemical synapses between them.

*Reviewer #2 (Recommendations for the authors):*

Patiño et al. developed a new rabies viral vector that contains both PSD95GFP (an excitatory postsynaptic marker) and SynPhRFP (a general presynaptic marker) to quantify what proportion of all excitatory presynaptic contacts on a starter cell are labeled by rabies virus-mediated retrograde monosynaptic tracing. They defined the labeling efficiency as the proportion of PSD95GFP puncta that are directly opposed to SynPhRFP puncta, and compared the labeling efficiency values among different dendritic positions (i.e. apical, proximal, and basal dendrites) or different neuronal types (i.e. L4 excitatory neurons, Sst+ interneurons, and Vip+ interneurons). They find no significant difference in the labeling efficiency depending on subcellular locations and neuronal types. They show that about 40% of first-order presynaptic excitatory contacts on a starter cell are labeled in their experimental conditions with careful control experiments. Overall, the experimental design is sound although additional validation is necessary. This work demonstrates "the labeling efficiency at synaptic levels" of rabies virus-mediated input tracing. However, there are several concerns in relation to their claim and the significance of this analysis.

1) The authors assume that all PSD95GFP puncta receive presynaptic contacts and the total number of PSD95GFP represents the total number of synapses on starter cells. This should be validated by examining the proportion of PSD95GFP puncta that associate with endogenous excitatory presynaptic markers such as VGLUT1.

2) The authors appear to use "the efficiency of spread" as a phrase equivalent to "the labeling efficiency". However, these are different; the efficiency of spread should be defined as the proportion of synapses through which rabies virus particles are actually retrogradely transported. Assuming that one input neuron makes five presynaptic contacts onto a starter cell, even if rabies virus particles pass through only one synapse among them, all five presynaptic contacts should be labeled. Thus, labeling efficiency depends on the efficiency of spread and the number of presynaptic contacts per input neuron. The efficiency of spread is equivalent to the labeling efficiency only when one input neuron forms one presynaptic contact with a starter cell.

3) There remain a number of questions about rabies virus-mediated input tracing, part of which are discussed by the authors. What percentage of all (excitatory or inhibitory) input cells are labeled? How large should the size of postsynaptic density be to allow for the retrograde spread of rabies virus particles? How many synapses does an input neuron form onto a starter cell? The number of synapses an input neuron forms depends on the subcellular locations of a starter cell or types of a starter cell. What presynaptic receptors are required for the transsynaptic spread of rabies viruses? Addressing these questions should be important for understanding the mechanism underlying rabies virus spread as well as more detailed connection properties of neural circuits. Unfortunately, the present study falls short of providing biological implications beyond "the labeling efficiency at synaptic levels".

1) In Figure 3, higher magnification images representing PSD95GFP puncta that associate with SypPhRFP puncta should be included.

2) It would be easier for readers to understand how the cross-over insensitive ATG-out (CIAO) AAV construct works if the authors could provide a concise explanation in the text.

*Reviewer #3 (Recommendations for the authors):*

The efficiency of rabies virus-mediated retrograde trans-synaptic tracing, one of the prevailing methods for deciphering cell-type specific neural connectivities, has long been in the debate yet has not definitively been addressed. To directly determine the efficiency, the authors took a simple and elegant strategy to visualize the excitatory post-synaptic structures with GFP-fused PSD95 and pre-synaptic structures with TagRFP-T-fused synaptophysin. They genetically modified the rabies genome to allow the simultaneous targetings of pre- and post-synaptic fluorescent markers to both the post-synaptic starter cells and their pre-synaptic partners. By high-resolution confocal microscopy, they showed along a dendrite of the starter post-synaptic neurons, the excitatory post-synaptic structures were visualized with GFP+ puncta, some of which were tightly paired with the RFP+ pre-synaptic puncta. After subtracting some background labeling caused regardless of trans-synaptic labeling, the authors revealed that, under their experimental conditions, approximately 42% of the post-synaptic puncta were coupled with the trans-synaptically labeled pre-synaptic puncta, hence providing the labeling efficiency of rabies-tracing. Importantly, the efficiency of rabies tracing was similar between excitatory and inhibitory starter cells, and also grossly invariant the along proximal vs. distal dendrites of the starter cells. These data provide useful basic properties of the rabies tracing method. However, the main conclusion could contain an overestimation because not all the labeled pre-synaptic structures were necessarily visualized from the starter cell of analysis. Overall, the present study at least estimated an upper limit of rabies tracing efficiency, which if carefully discussed should be useful for the community.

1) The conclusion can be accepted without modification if the total number of starter cells in their experimental condition was near one. However, the authors simply generated starter cells by co-infection of AAVs in a cell-type specific Cre driver mouse line, in which usually hundreds of starter cells are generated in the injection site. Given that the labeled pre-synaptic neurons can connect to multiple post-synaptic partners located nearby, not all the labeled pre-synaptic structures found along the labeled dendrites of a given starter cell were necessarily visualized from the exact starter cell of analysis. This property of neural circuits likely led to an overestimation of rabies-tracing efficiency reported in the present study. To characterize if this overestimation happened, the authors can first analyze the number of starter cells in their samples and then examine a correlation between the number of starter cells and the fraction of GFP/RFP coupling. Ideally, increasing the number of animal samples with a smaller number of total starter cells could provide a better estimate. Overall, the present study at least estimated an upper limit of rabies tracing efficiency, which if carefully discussed should be useful for the community.

2) Because GFP+ post-synaptic puncta were highly dense in the field of analysis, it is important to establish that the puncta detected along the Myc+ dendrites were mostly derived from the cell of analysis, not from adjacent uncharacterized cells. This can simply be achieved if the authors could analyze the density of GFP+ puncta along the Myc-positive but oG-negative dendrites as a negative control.

3) For reproducibility, it is important to report more details of viral constructions, in particular, RVdG-5PSD95eGFP-SynPhRFP, and describe resource availability. Because not all non-US researchers obtain rabies stock from the Salk Viral Core, depositing the corresponding plasmids (and most importantly the sequence information) to a more generally accessible distributor such as Addgene would be appreciated.

[Editors’ note: further revisions were suggested prior to acceptance, as described below.]

Thank you for resubmitting your work entitled "Detection of monosynaptic rabies spread at synaptic resolution reveals factors influencing tracing efficiency" for further consideration by *eLife*. Your revised article has been evaluated by Lu Chen (Senior Editor) and a Reviewing Editor.

The manuscript has been improved but there are some remaining issues that need to be addressed, as outlined below:

1) Please streamline the descriptions in the Results section "Relationships between…" (see comments from Reviewers #1 and #2 below). Please consider using new nomenclatures suggested by Reviewer #2 (all three reviewers agree on this during our discussion).

2) Please address Reviewer #1's critique #1 regarding the 4% probability of shared common inputs only applies to a pair of starter cells using suggested analysis and potentially new data.

3) Please address Reviewer #3's critique regarding the match between the title and data shown in the paper.

4) Please address all other critiques as much as you can.

*Reviewer #1 (Recommendations for the authors):*

The efficiency of rabies virus (RV)-mediated retrograde trans-synaptic tracing, a prominent technique for unraveling cell-type specific neural connectivities, has not yet been definitively addressed. To assess the labeling efficiency at the synaptic level and examine the relationship between the synaptic and cellular labeling, the authors employed rabies-encoding GFP-fused PSD95 and TagRFP-fused synaptophysin to simultaneously label both post-synaptic structures on the starter cells and their corresponding pre-synaptic counterparts. Following the subtraction of background labeling unrelated to trans-synaptic labeling, the authors revealed that, under their experimental conditions, approximately 40% of the post-synaptic puncta were coupled with the trans-synaptically labeled pre-synaptic puncta, thereby providing the labeling efficiency at the synapse level (synaptic efficiency, SE). Importantly, SE values were comparable between excitatory and inhibitory starter cells, as well as consistent along the proximal and distal dendrites of the starter cells. Lastly, the authors offer a theoretical framework to interpret SE and to potentially establish a connection with the cellular labeling efficiency of pre-synaptic neurons (Input probability, IP). While the manuscript encompasses significant technical and theoretical advancements, I noticed two major points that warrant careful consideration in interpreting the author's conclusions.

1. To analyze the associations between SE and IP, the authors appropriately introduced two factors. First, multiple starter neurons can share common input neurons, implying that the pre-synaptic structure of an observed starter neuron may be non-autonomously labeled from other starter cells. To characterize this effect, the authors conducted a thorough analysis of the existing literature and provided an estimated probability of 4% for two randomly selected cortical starter cells to share a common pre-synaptic neuron. Based on this estimate, they concluded that "this factor would be expected to result in SE being about 4% greater than IP". However, this conclusion holds only when analyzing samples with two starter cells. In reality, SE values are likely to be influenced by the number of starter cells (Ns) because, in a typical experiment involving Ns = 200-500 and given a 4% probability of shared common inputs in any pair of starter cells, nearly all labeled pre-synaptic neurons of the observed starter cell can be non-autonomously labeled from at least one of the other starter cells.

2. The authors also appropriately introduced the second point: multiple pre-synaptic structures provided by an input neuron can all be labeled if the RV passes through any one of them. I appreciate the author's attempt to suggest that this factor may have only a modest influence on the relationship between SE and IP (Figure 5). However, I am concerned that this factor has a more pronounced impact on the interpretation of the observed spatial organization of the pre-synaptic structures. Even in an extreme scenario where the RV passes through only the proximal (P) or distal (D) synapse, assuming that the pre-synaptic neurons evenly provide pre-synaptic structures along the PD axis, the result would be the comparable SE values between P and D synapses. The manuscript would benefit from a more meticulous discussion of this issue.

3. Considering my major point #1 above, I believe the manuscript would gain from the inclusion of an estimation of SE pertaining to a single starter cell, which is unlikely to be 0.4. The authors could begin by analyzing the number of starter cells in their samples and then perform a regression analysis to estimate the SE of a single starter cell. Increasing the number of animal samples with a smaller number of starter cells could contribute to obtaining a more accurate estimate. Furthermore, it would be advantageous to simulate the relationships between the number of starter cells (Ns) and SE, using assumed circuit structures and the efficiency of RV spread across a single synaptic contact (referred to as unitary efficiency, U).

4. Although I appreciate the authors' endeavors to estimate the relationships between SE and IP, the relevant text could benefit from a more concise writing style, aligning with the rest of the manuscript. For *eLife* readers with diverse backgrounds, the inclusion of schematic diagrams would be instructive.

*Reviewer #2 (Recommendations for the authors):*

G-deleted rabies virus has been widely used to trace monosynaptic inputs to specific populations of neurons in the mammalian brain since its first report by Callaway and colleagues in 2007. Despite its wide use, many fundamental properties of the method remain to be characterized. In this short report, Patino et al. addressed two important questions: (1) what the fraction of synapses does the rabies virus cross from a starter cell? (2) does this vary for synapses located at different distances to the soma? Answers to these questions should help researchers to interpret their data and thus will be of general interest to the neural circuit community.

To answer these questions, the authors produced a new rabies virus that expresses both a presynaptic marker tagged with RFP and a postsynaptic marker (for excitatory neurons) tagged with GFP, and simply counted within starter cells the fraction of GFP+ puncta that have an adjacent RFP punctum. Since most RFP puncta are a result of trans-synaptic transfer of rabies virus (the authors did a nice control of skipping G in one experiment), the authors can "quantify efficiency of spread at the synaptic level". The authors found about 35-40% of GFP+ puncta have RFP puncta next to them, and interestingly this number does not change whether these quantifications were performed close to the cell body or far from the cell body (thus answering question #2).

I appreciate the length the authors went to address my critiques of the original manuscript on the issue of distinguishing (1) fraction of synapses that are co-labeled in the starter cell, (2) fraction of input neurons that are labeled by rabies virus, and (3) average efficiency of rabies virus crossing an individual synapses. This will clarify to the readers what the authors have done and what kind of conclusions they can take away. However, I have several remaining critiques to the above issue that should be addressed by further textual revision.

1) I don't think the terms the authors used to describe the above three terms are the clearest. The authors used "synaptic efficiency" to describe (1), but it can easily be confused with (3). I recommend synapse proportion (SP) to parallel with input proportion (IP), or perhaps a bit better synapse fraction (SF) vs. input fraction (IF). Then (3) can be changed to "Unitary synapse efficiency (U)". These terms may be widely adopted in future publications on rabies tracing, so careful consideration is warranted.

2) The textual description in the Results section "Relationships between…" is often repetitive and needs to be streamlined. Specifically, I don't like the description of the results in the first paragraph ahead of all the analyses (and the authors have done these repetitively). For example, most of the first paragraph should be trimmed-just give the definition of the terms and move on to the actual "formal analysis." Summarize the results at the end.

3) The results of the formal analysis is based on the cortical circuit. It's important to emphasize that in circuits with denser connectivity, the relationship will be very different. As such, when these properties are discussed in other parts of the manuscript (abstract, introduction, discussion, etc.), please make sure to include the qualification phrase that these inferences were made from cortical circuits (e.g., last paragraph of Introduction).

*Reviewer #3 (Recommendations for the authors):*

1) Prior studies have quantified the proportion of PSD95 labeled postsynaptic densities that are opposed by presynaptic terminals that are also labeled with various presynaptic markers, including synaptophysin and VGLUT1 (Neuron. 2010 Nov 18; 68(4): 639-653. doi: 10.1016/j.neuron.2010.09.024). Essentially every PSD95 labeled terminal has a labeled presynaptic contact. Synaptophysin is a reliable marker of presynaptic terminals but VGLUT1 is not. We have modified the text in the introduction which now reads: "We designed a new genetically modified rabies virus that labels presynaptic terminals with synaptophysin-RFP (SynPhRFP) and excitatory postsynaptic densities with postsynaptic density-95-GFP (PSD95GFP). Because more than 99% of PSD95 postsynaptic puncta co-localize with a presynaptic terminal (Micheva et al., 2010), this construct allows us to quantify the proportion of excitatory synapses on a starter cell that have their corresponding input neuron labeled with rabies virus (defined here as synaptic efficiency [SE])."

Comment: I understand that "endogenous" PSD95 is a reliable postsynaptic marker in vivo. However, considering this is a new reagent and rabies viruses highly express transgenes, it is fair to experimentally show the colocalization rate of PSD95GFP expressed from rabies viruses and endogenous presynaptic markers (e.g. synaptophysin).

2) We thank the reviewer for this excellent point. We have been very careful not to equate efficiency of spread to labeling efficiency. As described in the above response to a similar comment from reviewer #1 we have added a new Results section, "Relationships between synaptic efficiency, input proportion, and unitary efficiency". Here the value that we formally define as unitary efficiency (U) is equivalent to what the reviewer terms "efficiency of spread", while our measured values of synaptic efficiency (SE) correspond to what the reviewer describes as "labeling efficiency". We show that there are complex relationships between the proportion of labeled presynaptic terminals (SE), efficiency of spread across individual synapses (U), the distributions of numbers of synaptic contacts between connected neuron pairs, and the proportion of input neurons labeled (IP).

Comment: I appreciate the authors' effort to estimate IP and U from SE and that the authors clearly discriminate SE and the spread efficiency in the text. The results are informative to some extent but not definitive because they rely on several assumptions such as "U is identical across synapses" and "there are two populations of input neurons that make the distinct numbers of presynaptic contacts per neuron onto starter cells", which are likely different from real situations. I am afraid that this analysis is not strong enough to claim the utility of the novel rabies vectors.

3) We would like to thank the reviewer for these thought-provoking questions. This is a rather disparate set of comments each of which relates to very different properties of either cortical circuit organization or rabies tracing. Some of these are known for limited aspects of cortical circuits while others remain unknown. With respect to rabies virus spread, some of these are known or can be inferred from published observations.

Below we further address each of the reviewer's specific questions.

Comment: There seems to be a misunderstanding of my previous review comment. I just listed examples of key questions about neural circuit organization and the mechanisms for the rabies virus spread, whose answers should bring important biological insights. I did not mean to request the authors to address all these questions.

4) We disagree with the stated characterization of our results: "Addressing these questions should be important for understanding the mechanism underlying rabies virus spread as well as more detailed connection properties of neural circuits. Unfortunately, the present study falls short of providing biological implications beyond "the labeling efficiency at synaptic levels". First, it was not a goal of our study to understand the mechanisms underlying the spread of rabies virus nor did we intend to reveal detailed connection properties of neural circuits. As stated in our Introduction, we were interested in identifying factors that influence the efficiency of transynaptic spread of rabies and we designed our experiments to address two such factors. These are the distance of synaptic contacts from the cell body and the type of postsynaptic neuron. Our data show that the proportions of labeled synaptic contacts are similar regardless of distance from the cell body and regardless of the postsynaptic cell types tested. These are quite important factors and given that the overall efficiency of spread can easily be increased or decreased by changing glycoprotein expression levels, these measures are more important than the overall measured efficiency in our experiments.

Comment: The title of the manuscript is now "Detection of monosynaptic rabies spread at synaptic resolution reveals factors influencing tracing efficiency". However, major achievements/findings of this study are that new rabies virus vectors that enable visualization of input synapses onto starter cells and estimation of the fraction of labeled presynaptic contacts are developed and that the proportions of labeled synaptic contacts are similar regardless of distance from the cell body and regardless of the postsynaptic cell types tested, as the authors state above. I am afraid that the authors do not provide "factors that influence (alter) tracing efficiency", which makes me feel that the title does not match with the data. If the scope of the manuscript is neither to understand the mechanisms underlying the spread of rabies virus nor to reveal detailed connection properties of neural circuits, can the authors clarify what kind of important questions could be addressed using this novel tool (the utility of the tool)?

---

## [Author Response]

[Editors’ note: the authors resubmitted a revised version of the paper for consideration. What follows is the authors’ response to the first round of review.]

Comments from all three reviewers are attached. The reviewers also had a robust discussionReviewer #1 (Recommendations for the authors):G-deleted rabies virus has been widely used to trace monosynaptic inputs to specific populations of neurons in the mammalian brain since its first report by Callaway and colleagues in 2007. Despite its wide use, many fundamental properties of the method remain to be characterized. In this short report, Patino et al. addressed two important questions: (1) what fraction of synapses does the rabies virus cross from a starter cell? (2) does this vary for synapses located at different distances to the soma? Answers to these questions should help researchers to interpret their data and thus will be of general interest to the neural circuit community.To answer these questions, the authors produced a new rabies virus that expresses both a presynaptic marker tagged with RFP and a postsynaptic marker (for excitatory neurons) tagged with GFP, and simply counted within starter cells the fraction of GFP+ puncta that have an adjacent RFP punctum. Since most RFP puncta are a result of the trans-synaptic transfer of rabies virus (the authors did a nice control of skipping G in one experiment), the authors can "quantify efficiency of spread at the synaptic level". The authors found about 35-40% of GFP+ puncta have RFP puncta next to them, and interestingly this number does not change whether these quantifications were performed close to the cell body or far from the cell body (thus answering question #2). However, the current version has a number of issues that should be addressed.1. While the authors conclude properly that under their experimental conditions, "35% to 40% of excitatory inputs to the examined cell types were labeled," many readers would mistakenly think that "rabies virus crosses 35-40% of synapses of the starter cells" from reading this paper. The latter is unlikely true for two reasons. First, if starter cell 1 transfers rabies virus to presynaptic neuron 1, which not only connects with starter cell 1 but also starter cell 2, then a pair of green/red puncta between starter cell 2 and presynaptic neuron 1 could result even if rabies virus never crossed the synapse between starter cell 2 and presynaptic neuron 1. Second, if starter cell 1 forms 5 synapses with presynaptic neuron 1, then it is possible that the rabies virus just crossed 1 of the 5 synapses (there may be some data supporting 5-10 synapses between synaptic pairs of cortical excitatory neurons), and the other 4 pairs of puncta are incidental. Thus, the actual fraction of synapses that the rabies virus crosses is likely substantially below 35-40%. While the first caveat can be overcome by doing experiments from single starter cells, the second caveat cannot be avoided. It is possible that the authors could estimate -based on cortical connectivity data (luckily V1 is one of the best studied in this regard, including serial EM reconstruction)-"what fraction of synapses does the rabies virus cross," which is a more fundamental question likely generalizable across cell types than "what fraction of excitatory inputs are labeled," which will vary greatly across circuits because of specific connection properties. At the very least, the authors should distinguish these two questions clearly, and state why they can differ, so that readers won't come away being misled.

We thank the reviewer for these excellent points. We have added a new section “Relationships between synaptic efficiency, input proportion, and unitary efficiency” to the results, which also includes a new figure (Figure 5). In this section we formally consider and quantitatively assess the impacts of both of the factors mentioned above (shared inputs and input neurons with multiple synaptic contact). This section along with a new methods section (“Calculations of synaptic efficiency (SE) and input proportion (IP)”) describes the derivation of formal equations that allow an evaluation of how the proportions of labeled presynaptic terminals in our experiments (SE) relate to the proportion of input neurons (IP) labeled and the probability (U) that rabies virus spreads across a single synaptic contact (unitary efficiency). Consideration of known properties of cortical circuits informs the range of variables that are input to the derived equations and the resulting values under different conditions are plotted in the figures. Based on these calculations we infer that our observed value of about 0.4 for SE likely corresponds to about 32% of input neurons labeled and a probability of spread across a single contact of 25%. Further, under unlikely, “worst-case scenario” conditions, percentage of input neurons labeled would be about 24%.

We believe that the addition of this material, as well as slight modifications in wording throughout the manuscript, will help to assure that readers will not incorrectly interpret our results and “mistakenly think that "rabies virus crosses 35-40% of synapses of the starter cells" from reading this paper.” For example, the introduction now summarize this part of the results (which is too long to fit in the Abstract) as: “Finally, we model the factors that influence the relationship between SE and the proportion of input neurons labeled (Input Probability, IP) and show that under plausible conditions the IP/SE ratio is about 0.75, so an SE of 40% corresponds to an IP of 30% of input neurons labeled. This analysis also allows an estimate of the probability of rabies spread across a single synaptic contact (unitary efficiency, U) which we estimate to be about 0.25 (25%) when SE=0.4.”

2. I am confused by Figure 2F, one of the most important panels of the entire paper that provides examples of GFP/RFP co-staining at high resolution. GFP marks postsynaptic density and thus should colocalize with the dendritic marker (magenta) and RFP marks presynaptic terminals of presynaptic neurons and should be adjacent but outside the magenta-positive area. But the third row (the positive example) shows the exact opposite. Red is within the magenta area, whereas green is adjacent to the magenta area. Can the authors clarify?

Regarding GFP and smFP_myc colocalization we would like to highlight that while not perfectly aligned the GFP puncta is indeed within the magenta positive region. We have provided zoomed in images below to highlight this point. Additionally, we have added additional images of synapse examples to figure 3 which further highlight this point.

Regarding overlap of the RFP presynaptic terminal with the magenta cytoplasmic marker, we want to highlight two points. One, the example images are a max intensity projection (MIP) reconstruction of multiple z-planes to enable 2D visualization of 3D structures. Therefore, structures in distinct z-planes may appear to co-localize in the flattened 2D image, for example if the presynaptic axon is above or below the spine. We have edited the manuscript to highlight this critical point by adding the following sentence to the methods sections “Images presented in manuscript are max intensity projection reconstructions to allow for 2D visualization of 3D structures'' and by adding “max intensity projection reconstruction” when describing example images. Of note, MIP was only done for manuscript visualization purposes and counting analysis was done one z-plane at a time as stated in the methods section. Second, although the resolution offered with Airyscan imaging is improved compared to confocal microscopy, it is not without resolution limits. It is therefore likely that structures may not perfectly align as expected, particularly in the axial dimension which has the poorest resolution. Indeed the size of a synaptic cleft (20-40nm) is outside of the resolution limit. Therefore, if the input axon is synapsing perpendicular to the x-y plane it would appear within the magenta region. Finally, we would like to point out that the synapse to the right of the yellow boxed region in Figure 2.F top panel as an example of “RFP should be adjacent but outside the magenta-positive area”. While some synapses do adhere to these expectations, for the reasons discussed above, not all do.

**Author response image 1. sa2fig1:** 

3. This brings an important issue of setting objective criteria for synaptic pairs: how much overlap, what's the maximum/minimum distance, etc, as this is the basis for the entire study. Indeed, what counts as a synaptic pair based on pre and postsynaptic markers in light microscopy is a debated issue. The authors are helped here by the sparse labeling of starter cells and presynaptic partners; still, I highly recommend that the authors state clearly the criteria in Methods.

We have added the following description to the methods section to address this point:

“Airyscan super-resolution imaging can resolve structures ~120 nm apart in the lateral dimension and ~350 nm apart in the axial dimension. The synaptic cleft size, or space between the pre and postsynaptic density, ranges from 20-30 nm, making it outside of the range of the resolution limit. Therefore, we required partial overlap, or colocalization, between pre and postsynaptic markers and did not allow for any distance between the two markers to consider them a synapse.”

4. The negative control of omitting oG in Figure 2 is very nice; I hope that the authors will do the same for tracing inputs to the two types of inhibitory neurons. These inhibitory neurons are also known to form gap junctions and it will be interesting to see whether the rabies virus would jump through gap junctions without oG, even if they are not known to form substantial chemical synapses between them.

Prior studies (Tang et al. 1999) have shown that rabies virus propagation occurs at chemical synapses but not via gap junctions. Therefore it is not expected that rabies would spread across gap junctions without oG. In addition, all of the components of the rabies virus and its genome are far larger than the small molecules that can spread through gap junctions. Furthermore as stated in the manuscript, Sst and Vip both exhibit very low levels of recurrent chemical synaptic connections, unlike L4 excitatory neurons, and therefore we did not find it necessary to perform similar controls as negligible numbers of direct connections between starter neurons are expected. Indeed, Sst and Vip neurons were chosen for experiments over Pvalb neurons, due to this unique property.

Reviewer #2 (Recommendations for the authors):Patiño et al. developed a new rabies viral vector that contains both PSD95GFP (an excitatory postsynaptic marker) and SynPhRFP (a general presynaptic marker) to quantify what proportion of all excitatory presynaptic contacts on a starter cell are labeled by rabies virus-mediated retrograde monosynaptic tracing. They defined the labeling efficiency as the proportion of PSD95GFP puncta that are directly opposed to SynPhRFP puncta, and compared the labeling efficiency values among different dendritic positions (i.e. apical, proximal, and basal dendrites) or different neuronal types (i.e. L4 excitatory neurons, Sst+ interneurons, and Vip+ interneurons). They find no significant difference in the labeling efficiency depending on subcellular locations and neuronal types. They show that about 40% of first-order presynaptic excitatory contacts on a starter cell are labeled in their experimental conditions with careful control experiments. Overall, the experimental design is sound although additional validation is necessary. This work demonstrates "the labeling efficiency at synaptic levels" of rabies virus-mediated input tracing. However, there are several concerns in relation to their claim and the significance of this analysis.1) The authors assume that all PSD95GFP puncta receive presynaptic contacts and the total number of PSD95GFP represents the total number of synapses on starter cells. This should be validated by examining the proportion of PSD95GFP puncta that associate with endogenous excitatory presynaptic markers such as VGLUT1.

Prior studies have quantified the proportion of PSD95 labeled postsynaptic densities that are apposed by presynaptic terminals that are also labeled with various presynaptic markers, including synaptophysin and VGLUT1 (Neuron. 2010 Nov 18; 68(4): 639–653. doi: 10.1016/j.neuron.2010.09.024). Essentially every PSD95 labeled terminal has a labeled presynaptic contact. Synaptophysin is a reliable marker of presynaptic terminals but VGLUT1 is not. We have modified the text in the introduction which now reads: “We designed a new genetically modified rabies virus that labels presynaptic terminals with synaptophysin-RFP (SynPhRFP) and excitatory postsynaptic densities with postsynaptic density-95-GFP (PSD95GFP). Because more than 99% of PSD95 postsynaptic puncta co-localize with a presynaptic terminal (Micheva et al., 2010), this construct allows us to quantify the proportion of excitatory synapses on a starter cell that have their corresponding input neuron labeled with rabies virus (defined here as synaptic efficiency [SE]).”

2) The authors appear to use "the efficiency of spread" as a phrase equivalent to "the labeling efficiency". However, these are different; the efficiency of spread should be defined as the proportion of synapses through which rabies virus particles are actually retrogradely transported. Assuming that one input neuron makes five presynaptic contacts onto a starter cell, even if rabies virus particles pass through only one synapse among them, all five presynaptic contacts should be labeled. Thus, labeling efficiency depends on the efficiency of spread and the number of presynaptic contacts per input neuron. The efficiency of spread is equivalent to the labeling efficiency only when one input neuron forms one presynaptic contact with a starter cell.

We thank the reviewer for this excellent point. We have been very careful not to equate efficiency of spread to labeling efficiency. As described in the above response to a similar comment from reviewer #1 we have added a new Results section, “Relationships between synaptic efficiency, input proportion, and unitary efficiency”. Here the value that we formally define as unitary efficiency (U) is equivalent to what the reviewer terms “efficiency of spread”, while our measured values of synaptic efficiency (SE) correspond to what the reviewer describes as “labeling efficiency”. We show that there are complex relationships between the proportion of labeled presynaptic terminals (SE), efficiency of spread across individual synapses (U), the distributions of numbers of synaptic contacts between connected neuron pairs, and the proportion of input neurons labeled (IP).

3) There remain a number of questions about rabies virus-mediated input tracing, part of which are discussed by the authors.

We would like to thank the reviewer for these thought-provoking questions. This is a rather disparate set of comments each of which relates to very different properties of either cortical circuit organization or rabies tracing. Some of these are known for limited aspects of cortical circuits while others remain unknown. With respect to rabies virus spread, some of these are known or can be inferred from published observations.

We disagree with the stated characterization of our results: “Addressing these questions should be important for understanding the mechanism underlying rabies virus spread as well as more detailed connection properties of neural circuits. Unfortunately, the present study falls short of providing biological implications beyond "the labeling efficiency at synaptic levels". First, it was not a goal of our study to understand the mechanisms underlying the spread of rabies virus nor did we intend to reveal detailed connection properties of neural circuits. As stated in our Introduction, we were interested in identifying factors that influence the efficiency of transynaptic spread of rabies and we designed our experiments to address two such factors. These are the distance of synaptic contacts from the cell body and the type of postsynaptic neuron. Our data show that the proportions of labeled synaptic contacts are similar regardless of distance from the cell body and regardless of the postsynaptic cell types tested. These are quite important factors and given that the overall efficiency of spread can easily be increased or decreased by changing glycoprotein expression levels, these measures are more important than the overall measured efficiency in our experiments.

Below we further address each of the reviewer’s specific questions.

What percentage of all (excitatory or inhibitory) input cells are labeled?

We quantified the spread of rabies virus across excitatory connections and not inhibitory connections. As described in the Discussion: “Furthermore, it is important to note that we only examined spread efficiency at excitatory synapses labeled with the excitatory postsynaptic density marker PSD-95, and did not quantify the efficiency of spread to inhibitory presynaptic inputs.”

We are able to estimate the percentage of excitatory input cells labeled based on our measured values of proportion of PSD95 puncta that are labeled. These estimates depend on many factors that are only partially known, but as detailed in the new Results section, we are able to estimate within a reasonably tight range.

How large should the size of postsynaptic density be to allow for the retrograde spread of rabies virus particles? The number of synapses an input neuron forms depends on the subcellular locations of a starter cell or types of a starter cell. What presynaptic receptors are required for the transsynaptic spread of rabies viruses? Addressing these questions should be important for understanding the mechanism underlying rabies virus spread as well as more detailed connection properties of neural circuits. Unfortunately, the present study falls short of providing biological implications beyond "the labeling efficiency at synaptic levels".

We combine these questions because the answers are interrelated. Despite extensive efforts over several decades to identify receptors for rabies virus uptake, there are not any known molecules that are required for the spread of rabies virus across synaptic contacts. Rabies virus shares this property with many other enveloped viruses such as Vesicular Stomatitis Virus. All available evidence indicates that the only requirement for spread is the presence of a synaptic contact, as rabies does not spread between neurons in vivo in the absence of synaptic contacts. This is a unique property of rabies, likely due to the lack of hetero-dimeric G-proteins that would allow both stabilization of the particle at the cell membrane and subsequent uptake. Instead the homo-dimeric rabies glycoprotein cannot be stabilized at the cell membrane and thus is susceptible to uptake only at synaptic contacts where the viral particle is larger than the synaptic cleft and is immediately stabilized at the presynaptic membrane when budding from a postsynaptic specialization. It is beyond the scope of this manuscript to include details about the mechanisms of rabies spread, about which we conduct no experiments. If only synaptic contact is required, it follows that the probability of rabies virus spread depends on the size of the synaptic contact, and since contact size is related to functional strength, it would follow that the probability of spread is also likely to be related to functional strength. Further, the probability that an input neuron would be labeled should also depend on the number of synaptic contacts it provides. This is an assumption of the formal model of spread and neuron labeling that we have added in the new Results section. We also consider these factors in our Discussion.

It would be of interest to experimentally measure the relationship between synapse size and probability of labeling. For example, this might be accomplished by combining rabies labeling with electron microscopy, which is the most sensitive and accurate method to study questions relevant to synaptic size. This would require the development of novel rabies constructs that are compatible with EM. The Discussion now states: “Indeed, one limitation of our experimental design is that our light microscopy images do not allow accurate measurement of postsynaptic density size, which would require electron microscopy.”

How many synapses does an input neuron form onto a starter cell?

There is not a singular answer to this question because for known connections the number varies depending on the pre- and postsynaptic cell types and there is also diversity within those populations. In the Results section we now consider known values for the small numbers of excitatory cortical connections onto excitatory cortical neurons that have been quantified and assess how those impact the relationships between our measured values of proportions of postsynaptic sites that are labeled and other values of interest.

1) In Figure 3, higher magnification images representing PSD95GFP puncta that associate with SypPhRFP puncta should be included.

Representative higher magnification images have been added to Figure 3.

2) It would be easier for readers to understand how the cross-over insensitive ATG-out (CIAO) AAV construct works if the authors could provide a concise explanation in the text.

We have added the following to the main text.

“CIAO constructs have the ATG codon placed outside of the loxp mutant pairs loxp66/71 sites to ensure the gene coding region is out of frame with the ATG start signal in the absence of Cre-mediated recombination”

Reviewer #3 (Recommendations for the authors):The efficiency of rabies virus-mediated retrograde trans-synaptic tracing, one of the prevailing methods for deciphering cell-type specific neural connectivities, has long been in the debate yet has not definitively been addressed. To directly determine the efficiency, the authors took a simple and elegant strategy to visualize the excitatory post-synaptic structures with GFP-fused PSD95 and pre-synaptic structures with TagRFP-T-fused synaptophysin. They genetically modified the rabies genome to allow the simultaneous targetings of pre- and post-synaptic fluorescent markers to both the post-synaptic starter cells and their pre-synaptic partners. By high-resolution confocal microscopy, they showed along a dendrite of the starter post-synaptic neurons, the excitatory post-synaptic structures were visualized with GFP+ puncta, some of which were tightly paired with the RFP+ pre-synaptic puncta. After subtracting some background labeling caused regardless of trans-synaptic labeling, the authors revealed that, under their experimental conditions, approximately 42% of the post-synaptic puncta were coupled with the trans-synaptically labeled pre-synaptic puncta, hence providing the labeling efficiency of rabies-tracing. Importantly, the efficiency of rabies tracing was similar between excitatory and inhibitory starter cells, and also grossly invariant the along proximal vs. distal dendrites of the starter cells. These data provide useful basic properties of the rabies tracing method. However, the main conclusion could contain an overestimation because not all the labeled pre-synaptic structures were necessarily visualized from the starter cell of analysis. Overall, the present study at least estimated an upper limit of rabies tracing efficiency, which if carefully discussed should be useful for the community.

We thank the reviewer for these comments

We would like to clarify that our results do not estimate “an upper limit of rabies tracing efficiency”. Instead, for the particular reagents and tracing conditions used, our measures of the proportion of postsynaptic densities with labeled presynaptic terminals provides an upper limit on the proportion of input neurons labeled as well as an upper limit on the probability that rabies spreads across a single synaptic contact. But, as has been shown in prior publications, these values can be readily increased or decreased by changing tracing conditions, such as the titer of rabies virus used, the strain of rabies virus used, the expression levels or strain of rabies glycoprotein, or survival time after rabies injections. These issues are addressed in the manuscript Introduction and Discussion.

1) The conclusion can be accepted without modification if the total number of starter cells in their experimental condition was near one. However, the authors simply generated starter cells by co-infection of AAVs in a cell-type specific Cre driver mouse line, in which usually hundreds of starter cells are generated in the injection site. Given that the labeled pre-synaptic neurons can connect to multiple post-synaptic partners located nearby, not all the labeled pre-synaptic structures found along the labeled dendrites of a given starter cell were necessarily visualized from the exact starter cell of analysis. This property of neural circuits likely led to an overestimation of rabies-tracing efficiency reported in the present study. To characterize if this overestimation happened, the authors can first analyze the number of starter cells in their samples and then examine a correlation between the number of starter cells and the fraction of GFP/RFP coupling. Ideally, increasing the number of animal samples with a smaller number of total starter cells could provide a better estimate. Overall, the present study at least estimated an upper limit of rabies tracing efficiency, which if carefully discussed should be useful for the community.

We thank the reviewer for highlighting the importance of addressing limitations of distinguishing between labeled presynaptic terminals that are the result of direct rabies infection versus indirect routes. As described in more detail in response to the other two reviews (above), we have added a new Results section “Relationships between synaptic efficiency, input proportion, and unitary efficiency” to formally, and quantitatively assess these issues. We note that the approach suggested by the reviewer would still fail to reveal precise values for efficiency of spread across a synaptic contact or the proportion of input neurons labeled. For example, even if there were only one starter cell and every labeled input cell were counted, the expected fraction of labeled postsynaptic densities would still depend on unknown values, including the total number of presynaptic neurons and the numbers of synaptic contacts provided by the various input neurons. The reasons for this should be apparent from the new Results section.

2) Because GFP+ post-synaptic puncta were highly dense in the field of analysis, it is important to establish that the puncta detected along the Myc+ dendrites were mostly derived from the cell of analysis, not from adjacent uncharacterized cells. This can simply be achieved if the authors could analyze the density of GFP+ puncta along the Myc-positive but oG-negative dendrites as a negative control.

We apologize to the reviewer for being unclear about the analysis methods. To clarify, due to the abundance of synapses across a 50m coronal section, GFP+ post-synaptic puncta were counted manually one z-plane at a time (with each z-step spanning 0.5 m) and confirmed to be colocalized with the Myc+ dendrite from the starter cell being analyzed. We have added the following text to the methods section to clarify this point. “Due to the abundance of synapses across a 50m coronal section, puncta were counted manually one z-plane at a time. Images presented in manuscript are max intensity projection reconstructions to allow for 2D visualization of 3D structures.” While puncta from uncharacterized input cells could be near the dendrite being analyzed they would not be expected to colocalize with the Myc+ dendrite when being counted one z-plane at a time. To address the second point, analyzing the density of GFP+ puncta along the Myc-positive but oG-negative dendrites would not serve as a negative control as these neurons are also infected with rabies and express their own GFP+ puncta; in fact such neurons were analyzed in control experiments that omitted the AAV-FLEX-H2BBFP-oG virus (Figure 2).

3) For reproducibility, it is important to report more details of viral constructions, in particular, RVdG-5PSD95eGFP-SynPhRFP, and describe resource availability. Because not all non-US researchers obtain rabies stock from the Salk Viral Core, depositing the corresponding plasmids (and most importantly the sequence information) to a more generally accessible distributor such as Addgene would be appreciated.

In response to the reviewer's concerns about reproducibility, we have begun the process to deposit the plasmid and sequence information for the RVdG-5PSD95eGFP-SynPhRFP viral construct at Addgene to make it more widely available to researchers. The plasmid sequence has also been included as a Supplementary file in the manuscript.

[Editors’ note: what follows is the authors’ response to the second round of review.]

The manuscript has been improved but there are some remaining issues that need to be addressed, as outlined below:1) Please streamline the descriptions in the Results section "Relationships between…" (see comments from Reviewers #1 and #2 below). Please consider using new nomenclatures suggested by Reviewer #2 (all three reviewers agree on this during our discussion).

We thank the reviewers and editor for these suggestions, which we believe have led to a more accessible and clearer description of our findings.

We have now moved most of the text that was in the "Relationships between…" Results section to the Materials and methods section. The remaining text in that section is now a streamlined summary, as suggested. Note that much of the related text that is now in the Methods section has been modified, both in response to issue #2 (below) and due to availability of new data on the numbers of synapses made per connection for excitatory cortical circuits as well as the rate of shared common input.

We have adopted new nomenclature suggested by Reviewer #2. Synaptic Efficiency (SE) is replaced by Synaptic Fraction (SF). Input proportion (IP) is replaced by Input Fraction (IF). Unitary Efficiency (U) is now defined with the term Unitary Synaptic Efficiency (U).

2) Please address Reviewer #1's critique #1 regarding the 4% probability of shared common inputs only applies to a pair of starter cells using suggested analysis and potentially new data.

We thank reviewer #1 for correctly pointing out that the proportions of synapses labeled due to shared common input will depend not only on the amount of shared input, but also on the numbers of starter cells that potentially receive common input. To address this issue we have collected and analyzed new data. We have analyzed data from the MiCRONS electron microscopy (EM) dataset to directly assess that rate with which layer 4 neurons in adult mouse visual cortex share common input and how this relates to the distance between neurons. And we have also collected new data and analyzed the density and spatial distributions of Creexpressing neurons that can potentially serve as starter cells in our experiments using the Nr5a1-Cre mouse line.

We used these two datasets to allow quantification of both the numbers of starter cells that might share common input and the proportion of our observed synaptic labeling that might be due to indirect labeling from shared inputs rather than direct spread of rabies across the labeled synaptic contact. These results and detailed analysis can be found in the revised Methods section. (As noted above, this description is in the Methods section in order to streamline the description in the Results, as suggested.)

3) Please address Reviewer #3's critique regarding the match between the title and data shown in the paper.

We have modified the title, which is now: “Postsynaptic cell type and synaptic distance do not determine efficiency of monosynaptic rabies virus spread measured at synaptic resolution”.

There appears to have been some confusion about what exactly are the “factors” that we were referring to in the previous title: “Detection of monosynaptic rabies spread at synaptic resolution reveals factors influencing tracing efficiency.” The reviewer stated: “I am afraid that the authors do not provide "factors that influence (alter) tracing efficiency", But in the immediately preceding sentence the reviewer lists exactly the same factors that we were referring to in the former title. “However, major achievements/findings of this study are that new rabies virus vectors that enable visualization of input synapses onto starter cells and estimation of the fraction of labeled presynaptic contacts are developed and that the proportions of labeled synaptic contacts are similar regardless of distance from the cell body and regardless of the postsynaptic cell types tested …”

We believe that the new title now clearly identifies the “factors” – postsynaptic cell type and synaptic distance – which our experiments have evaluated with respect to their influences on the efficiency of rabies labeling.

4) Please address all other critiques as much as you can.

We address all other critiques in the context of the reviewer’s comments below.

Reviewer #1 (Recommendations for the authors):The efficiency of rabies virus (RV)-mediated retrograde trans-synaptic tracing, a prominent technique for unraveling cell-type specific neural connectivities, has not yet been definitively addressed. To assess the labeling efficiency at the synaptic level and examine the relationship between the synaptic and cellular labeling, the authors employed rabies-encoding GFP-fused PSD95 and TagRFP-fused synaptophysin to simultaneously label both post-synaptic structures on the starter cells and their corresponding pre-synaptic counterparts. Following the subtraction of background labeling unrelated to trans-synaptic labeling, the authors revealed that, under their experimental conditions, approximately 40% of the post-synaptic puncta were coupled with the trans-synaptically labeled pre-synaptic puncta, thereby providing the labeling efficiency at the synapse level (synaptic efficiency, SE). Importantly, SE values were comparable between excitatory and inhibitory starter cells, as well as consistent along the proximal and distal dendrites of the starter cells. Lastly, the authors offer a theoretical framework to interpret SE and to potentially establish a connection with the cellular labeling efficiency of pre-synaptic neurons (Input probability, IP). While the manuscript encompasses significant technical and theoretical advancements, I noticed two major points that warrant careful consideration in interpreting the author's conclusions.

We thank the reviewer for noting the significance of the advancements provided by our work.

1. To analyze the associations between SE and IP, the authors appropriately introduced two factors. First, multiple starter neurons can share common input neurons, implying that the pre-synaptic structure of an observed starter neuron may be non-autonomously labeled from other starter cells. To characterize this effect, the authors conducted a thorough analysis of the existing literature and provided an estimated probability of 4% for two randomly selected cortical starter cells to share a common pre-synaptic neuron. Based on this estimate, they concluded that "this factor would be expected to result in SE being about 4% greater than IP". However, this conclusion holds only when analyzing samples with two starter cells. In reality, SE values are likely to be influenced by the number of starter cells (Ns) because, in a typical experiment involving Ns = 200-500 and given a 4% probability of shared common inputs in any pair of starter cells, nearly all labeled pre-synaptic neurons of the observed starter cell can be non-autonomously labeled from at least one of the other starter cells.

We thank the reviewer for pointing out the importance of considering the numbers of starter neurons that can potentially provide common input, which we failed to address in our consideration. If the numbers of such neurons were in fact 200-500 and each of those were to provide 4% common inputs this would indeed be a major problem. Such numbers would probably cause nearly every synapse to be labeled, far more than the ~40% that we observed. The actual numbers of starter cells are likely to be much less than 200, but more importantly the probability of sharing common input is highly dependent on the distance between starter neurons. This is because the rate of connections in cortex falls off quite quickly with distance. Thus, a consideration of the spacing between potential starter cells is also required.

To allow us to incorporate a consideration of not only the numbers of starter cells but also their distances from each other, we conducted a quantitative analysis of the distributions of Creexpressing neurons in the nr5a1-Cre mice. We crossed the nr5a1-Cre mice with tdTomato reporter mice and then marked the locations of every TdTomato-expressing neuron. We then analyzed their spatial distributions and incorporated these data into a new estimate of the potential impact of common inputs on the measured fraction of synapses labeled for each starter neuron (Synaptic Fraction, SF). We also gained access to new data and analyses of EM data from the MicRonS data set which densely reconstructs connections within a volume of adult mouse visual cortex. These data allowed us to directly assess the proportion of shared input between layer 4 excitatory neurons and how that changes with distance between layer 4 cells. We believe this allows for a far more accurate measure than the data that we previously used, which was based on recordings from neonatal brain slices where connectivity might differ from than in the adult.

A detailed description of these data, analyses and methods are found in the revised Methods section. We have also added an author who conducted these analyses and generated the related figures. The new analysis indicates that, at most, about 22% of the measured SF is due to shared input. Note that this is likely to be an overestimate because it assumes that every Cre-expressing neuron is a starter cell, but in our experience using similar reagents only about half of Cre+ neurons become starter cells. For our measured SF values of 40% this corresponds to 31% of input neurons being labeled (IF=0.31).

2. The authors also appropriately introduced the second point: multiple pre-synaptic structures provided by an input neuron can all be labeled if the RV passes through any one of them. I appreciate the author's attempt to suggest that this factor may have only a modest influence on the relationship between SE and IP (Figure 5). However, I am concerned that this factor has a more pronounced impact on the interpretation of the observed spatial organization of the pre-synaptic structures. Even in an extreme scenario where the RV passes through only the proximal (P) or distal (D) synapse, assuming that the pre-synaptic neurons evenly provide pre-synaptic structures along the PD axis, the result would be the comparable SE values between P and D synapses. The manuscript would benefit from a more meticulous discussion of this issue.

We thank the reviewer for pointing out that multiple contacts from a given starter neuron could theoretically influence the observed locations of labeled synapses independently from the location of the synapses across which the rabies virus had spread.

The potential magnitude of such an effect depends on the proportion of connected neurons that actually make multiple synaptic contacts onto a starter cells. As noted above, we have added new data based on EM reconstructions and analysis of nearly 77,000 adult cortical connections. These data show that the overwhelming majority of cortical connections involve only a single synaptic contact. This has resulted in far lower estimates of the impact of multiple contacts relative to estimates based on data from neonatal brain slices.

To directly address this issue we have added the following text to the Discussion:

“The same factors that we have considered for their impact on the relationships between measured SF, IF, and U are also important to consider with respect to the spatial locations of labeled synapses. This is because synaptic labeling resulting from either shared inputs or multiple synaptic contacts from a single input neuron are not the result of the spread of rabies across a synaptic contact. Our analyses show that the proportions of labeled synapses arising from these mechanisms is small relative to the variability and proportions of observed SF at distal versus proximal synapses (Figure 3G, 4F).”

We have also added new text to the Results pointing out that a portion of the synaptic locations measured could be due to indirect labeling and might not reflect the locations where the virus spread. The Results now state: “We found no significant differences between synaptic fraction at proximal regions of apical dendrites compared to the distal regions as determined by PSD95GFP and SynPhRFP puncta colocalization (49.14 ± 3.49% versus 57 ± 5.43% respectively, Wilcoxon rank-sum test, p = 0.34; n = 9 neurons across 3 mice, Figure 3G). We also compared synaptic fraction at basal dendrites versus apical dendrites and observed no significant difference (44.52 ± 3.96% versus 53.05 ± 3.99% respectively, Wilcoxon rank-sum test, p = 0.22; n = 9 neurons across 3 mice and n = 6 neurons across 3 mice, Figure 3G). Note that a fraction of the observed synaptic labeling might occur through indirect pathways rather than direct spread of rabies virus at the observed synapse (see Below). This fraction is small relative to the variability in our measured synaptic fraction at distal versus proximal or apical versus basal dendrites.”

3. Considering my major point #1 above, I believe the manuscript would gain from the inclusion of an estimation of SE pertaining to a single starter cell, which is unlikely to be 0.4. The authors could begin by analyzing the number of starter cells in their samples and then perform a regression analysis to estimate the SE of a single starter cell. Increasing the number of animal samples with a smaller number of starter cells could contribute to obtaining a more accurate estimate. Furthermore, it would be advantageous to simulate the relationships between the number of starter cells (Ns) and SE, using assumed circuit structures and the efficiency of RV spread across a single synaptic contact (referred to as unitary efficiency, U).

We thank the reviewer for this suggestion. While we agree that the assessment suggested by the reviewer could theoretically provide some useful insight, such studies would in practice present technical difficulties that would likely make the results uninterpretable. It has been well established that the efficiency of rabies tracing is highly dependent on both the expression levels of rabies glycoprotein and the numbers of rabies viral particles initially entering the starter cells (See JNS Review by Luo and Callaway). To reduce the numbers of starter cells, it would be necessary to reduce the titers of AAV helper virus used or to reduce the titer of the RV used. Either of these changes would result in reduced tracing efficiency and skew the results from regression analysis. Other alternatives, such as single cell electroporation of starter cells, would also result in changes to glycoprotein expression or rabies infection that would preclude direct comparisons across experiments with different numbers of starter cells.

We note that the equations that we have derived for relationships between U and SF (formerly SE) show that this is independent of the number of starter cells. The numbers of starter cells only effects SF with respect to common inputs, not the configuration of “assumed circuit structures”. We also expect that the reviewer’s impression about the expected SF that might be observed for a single starter cell will change in view of the new data we have provided about: (1) proportion of shared inputs between layer 4 neurons and dependence on distance; (2) distributions of Cre-expressing layer 4 neurons in nr5a1-Cre mice; and (3) numbers of synaptic contacts per connected pair.

4. Although I appreciate the authors' endeavors to estimate the relationships between SE and IP, the relevant text could benefit from a more concise writing style, aligning with the rest of the manuscript. For eLife readers with diverse backgrounds, the inclusion of schematic diagrams would be instructive.

We have replaced the text in the Results with a much more concise description and moved details to Methods. In view of the reduced complexity of the likely circuit configurations indicated by the new EM data we have opted not to add a schematic diagram.

Reviewer #2 (Recommendations for the authors):G-deleted rabies virus has been widely used to trace monosynaptic inputs to specific populations of neurons in the mammalian brain since its first report by Callaway and colleagues in 2007. Despite its wide use, many fundamental properties of the method remain to be characterized. In this short report, Patino et al. addressed two important questions: (1) what the fraction of synapses does the rabies virus cross from a starter cell? (2) does this vary for synapses located at different distances to the soma? Answers to these questions should help researchers to interpret their data and thus will be of general interest to the neural circuit community.To answer these questions, the authors produced a new rabies virus that expresses both a presynaptic marker tagged with RFP and a postsynaptic marker (for excitatory neurons) tagged with GFP, and simply counted within starter cells the fraction of GFP+ puncta that have an adjacent RFP punctum. Since most RFP puncta are a result of trans-synaptic transfer of rabies virus (the authors did a nice control of skipping G in one experiment), the authors can "quantify efficiency of spread at the synaptic level". The authors found about 35-40% of GFP+ puncta have RFP puncta next to them, and interestingly this number does not change whether these quantifications were performed close to the cell body or far from the cell body (thus answering question #2).I appreciate the length the authors went to address my critiques of the original manuscript on the issue of distinguishing (1) fraction of synapses that are co-labeled in the starter cell, (2) fraction of input neurons that are labeled by rabies virus, and (3) average efficiency of rabies virus crossing an individual synapses. This will clarify to the readers what the authors have done and what kind of conclusions they can take away. However, I have several remaining critiques to the above issue that should be addressed by further textual revision.1) I don't think the terms the authors used to describe the above three terms are the clearest. The authors used "synaptic efficiency" to describe (1), but it can easily be confused with (3). I recommend synapse proportion (SP) to parallel with input proportion (IP), or perhaps a bit better synapse fraction (SF) vs. input fraction (IF). Then (3) can be changed to "Unitary synapse efficiency (U)". These terms may be widely adopted in future publications on rabies tracing, so careful consideration is warranted.

We thank the reviewer for this excellent suggestion. We have adopted the suggested nomenclature.

2) The textual description in the Results section "Relationships between…" is often repetitive and needs to be streamlined. Specifically, I don't like the description of the results in the first paragraph ahead of all the analyses (and the authors have done these repetitively). For example, most of the first paragraph should be trimmed-just give the definition of the terms and move on to the actual "formal analysis." Summarize the results at the end.

We have replaced the text in the Results section with a more concise description and reserved details for the Methods section.

3) The results of the formal analysis is based on the cortical circuit. It's important to emphasize that in circuits with denser connectivity, the relationship will be very different. As such, when these properties are discussed in other parts of the manuscript (abstract, introduction, discussion, etc.), please make sure to include the qualification phrase that these inferences were made from cortical circuits (e.g., last paragraph of Introduction).

We have made the suggested edits in all relevant cases that we have found. The last paragraph of the Introduction now explicitly refers to “our results in mouse visual cortex.” And in the discussion: “These equations can be used to assess circuit configurations that are likely to be different in other brain areas than for the cortical circuits on which we have focused.”

Reviewer #3 (Recommendations for the authors):1) Prior studies have quantified the proportion of PSD95 labeled postsynaptic densities that are opposed by presynaptic terminals that are also labeled with various presynaptic markers, including synaptophysin and VGLUT1 (Neuron. 2010 Nov 18; 68(4): 639-653. doi: 10.1016/j.neuron.2010.09.024). Essentially every PSD95 labeled terminal has a labeled presynaptic contact. Synaptophysin is a reliable marker of presynaptic terminals but VGLUT1 is not. We have modified the text in the introduction which now reads: "We designed a new genetically modified rabies virus that labels presynaptic terminals with synaptophysin-RFP (SynPhRFP) and excitatory postsynaptic densities with postsynaptic density-95-GFP (PSD95GFP). Because more than 99% of PSD95 postsynaptic puncta co-localize with a presynaptic terminal (Micheva et al., 2010), this construct allows us to quantify the proportion of excitatory synapses on a starter cell that have their corresponding input neuron labeled with rabies virus (defined here as synaptic efficiency [SE])."Comment: I understand that "endogenous" PSD95 is a reliable postsynaptic marker in vivo. However, considering this is a new reagent and rabies viruses highly express transgenes, it is fair to experimentally show the colocalization rate of PSD95GFP expressed from rabies viruses and endogenous presynaptic markers (e.g. synaptophysin).

The text quoted above was in response to a previous comment from this reviewer which stated: “The authors assume that all PSD95GFP puncta receive presynaptic contacts and the total number of PSD95GFP represents the total number of synapses on starter cells. This should be validated by examining the proportion of PSD95GFP puncta that associate with endogenous excitatory presynaptic markers such as VGLUT1.”

Our response to that comment was intended to clarify that VGLUT1 would not be a good choice, since it is not a reliable marker of presynaptic terminals. The text that we added to the Introduction emphasizes the fact that it has already been established that essentially all PSD95 puncta do in fact receive presynaptic contacts. Furthermore, we show in Figure 1 that the PSD95GFP puncta on rabies infected neurons colocalize with PSD95 antibody staining, precluding the possibility that the PSD95GFP from the rabies genome might somehow fail to label some PSDs. The reviewer’s new comments seem to be suggesting that perhaps through some unknown mechanism, the over-expression of PSD95GFP will result in the formation of new puncta that do not have a presynaptic input. Although this seems very unlikely, if this were to happen it would result in an underestimate of the rate at which rabies virus spreads. We have added new text to the discussion that states: “It should also be noted that the locations and numbers of postsynaptic sites was assessed based on expression of PSD-95-GFP from the rabies virus. If this overexpression were to somehow result in new puncta that do not have a presynaptic input this could result in an underestimate of proportion of actual PSDs labeled.”

2) We thank the reviewer for this excellent point. We have been very careful not to equate efficiency of spread to labeling efficiency. As described in the above response to a similar comment from reviewer #1 we have added a new Results section, "Relationships between synaptic efficiency, input proportion, and unitary efficiency". Here the value that we formally define as unitary efficiency (U) is equivalent to what the reviewer terms "efficiency of spread", while our measured values of synaptic efficiency (SE) correspond to what the reviewer describes as "labeling efficiency". We show that there are complex relationships between the proportion of labeled presynaptic terminals (SE), efficiency of spread across individual synapses (U), the distributions of numbers of synaptic contacts between connected neuron pairs, and the proportion of input neurons labeled (IP).Comment: I appreciate the authors' effort to estimate IP and U from SE and that the authors clearly discriminate SE and the spread efficiency in the text. The results are informative to some extent but not definitive because they rely on several assumptions such as "U is identical across synapses" and "there are two populations of input neurons that make the distinct numbers of presynaptic contacts per neuron onto starter cells", which are likely different from real situations. I am afraid that this analysis is not strong enough to claim the utility of the novel rabies vectors.

We would like to once again reinforce the fact that our primary aim in these experiments was not to determine what proportion of synapses are labeled but rather to determine whether the probability of labeling synapses depended on dendritic location or postsynaptic cell type. We show that these factors do not have a large impact. The results show that we were able to compare labeling at distal versus proximal sites and onto different cell types, as intended.

We have also included extensive discussion about the expected impact of synapse size and the likelihood that this is related to the efficiency of spread across a single synapse. We do not claim the “U is identical across synapses”, but this is a simplifying assumption in our model that allows the impact of the distributions of synapses per connection to be evaluated independently from potential complications related to synapse size.

We have also provided new data on the distribution of the numbers of synapses formed between connected excitatory cortical neurons. These data show that 92% of the connections involve only one synapse and that 99% of connections have only 1 or 2 synapses. Thus, 99% of connections can be accounted for by a simulation that considers just two populations of input neurons, one population with a single synapse (92% of connections) and a second population with two synapses per connection. This model is considered in Figures 6C and D. In Figures 6A and B, we also consider an extreme model (one population with one synapse and the other with three synapses) which shows that the impact of these parameters remains small even if the population of connections with three synapses were 10-fold larger than observed experimentally.

3) We would like to thank the reviewer for these thought-provoking questions. This is a rather disparate set of comments each of which relates to very different properties of either cortical circuit organization or rabies tracing. Some of these are known for limited aspects of cortical circuits while others remain unknown. With respect to rabies virus spread, some of these are known or can be inferred from published observations.Below we further address each of the reviewer's specific questions.Comment: There seems to be a misunderstanding of my previous review comment. I just listed examples of key questions about neural circuit organization and the mechanisms for the rabies virus spread, whose answers should bring important biological insights. I did not mean to request the authors to address all these questions.4) We disagree with the stated characterization of our results: "Addressing these questions should be important for understanding the mechanism underlying rabies virus spread as well as more detailed connection properties of neural circuits. Unfortunately, the present study falls short of providing biological implications beyond "the labeling efficiency at synaptic levels". First, it was not a goal of our study to understand the mechanisms underlying the spread of rabies virus nor did we intend to reveal detailed connection properties of neural circuits. As stated in our Introduction, we were interested in identifying factors that influence the efficiency of transynaptic spread of rabies and we designed our experiments to address two such factors. These are the distance of synaptic contacts from the cell body and the type of postsynaptic neuron. Our data show that the proportions of labeled synaptic contacts are similar regardless of distance from the cell body and regardless of the postsynaptic cell types tested. These are quite important factors and given that the overall efficiency of spread can easily be increased or decreased by changing glycoprotein expression levels, these measures are more important than the overall measured efficiency in our experiments.Comment: The title of the manuscript is now "Detection of monosynaptic rabies spread at synaptic resolution reveals factors influencing tracing efficiency". However, major achievements/findings of this study are that new rabies virus vectors that enable visualization of input synapses onto starter cells and estimation of the fraction of labeled presynaptic contacts are developed and that the proportions of labeled synaptic contacts are similar regardless of distance from the cell body and regardless of the postsynaptic cell types tested, as the authors state above. I am afraid that the authors do not provide "factors that influence (alter) tracing efficiency", which makes me feel that the title does not match with the data. If the scope of the manuscript is neither to understand the mechanisms underlying the spread of rabies virus nor to reveal detailed connection properties of neural circuits, can the authors clarify what kind of important questions could be addressed using this novel tool (the utility of the tool)?

We have modified the title, which is now: “Postsynaptic cell type and synaptic distance do not determine efficiency of monosynaptic rabies virus spread measured at synaptic resolution”.

There appears to have been some confusion about what exactly are the “factors” that we were referring to in the previous title: “Detection of monosynaptic rabies spread at synaptic resolution reveals factors influencing tracing efficiency.” The reviewer stated: “I am afraid that the authors do not provide "factors that influence (alter) tracing efficiency", But in the immediately preceding sentence the reviewer lists exactly the same factors that we were referring to in the former title. “However, major achievements/findings of this study are that new rabies virus vectors that enable visualization of input synapses onto starter cells and estimation of the fraction of labeled presynaptic contacts are developed and that the proportions of labeled synaptic contacts are similar regardless of distance from the cell body and regardless of the postsynaptic cell types tested …”

We believe that the new title now clearly identifies the “factors” – postsynaptic cell type and synaptic distance – which our experiments have evaluated with respect to their influences on the efficiency of rabies labeling.